# Learnable Context-Aware Attention Mask for Multimodal Transformers

## Abstract

The Self-Attention mechanism in Transformer models has shown great success across many domains, but its effectiveness can diminish in complex settings, such as multimodal tasks. This is due to the varying token granularity and the high computational cost of processing long sequences. To overcome these limitations, we propose the Learnable Context-Aware Attention Mask (LCAAM), a novel method that globally adjusts attention maps to prioritize the most important tokens in a sequence. Our approach integrates LCAAM into a BERT-like Transformer network, enhancing the Self-Attention mechanism by capturing token relationships while accounting for their contextual relevance. Additionally, we extend LCAAM to a multi-layer framework, enabling it to capture diverse information across the layers of the Transformer. Extensive experiments on datasets including MADv2, QVHighlights, ImageNet-1K, and MSRVTT demonstrate that LCAAM improves model performance while reducing redundant computations. This innovation offers a significant improvement in tackling complex tasks, such as movie understanding.

## 1 Introduction

The evolution of deep learning has empowered us to navigate increasingly complex scenarios, many of which require digesting information from diverse sources, such as videos, images, audio, and text. One such scenario lies in understanding movie scenes (Soldan et al., 2022; Han et al., 2023b;c; Barrios et al., 2023; Rohrbach et al., 2015; Xiao et al., 2022; Islam et al., 2023; Chen et al., 2023), where models aim to extract meaningful insights from multiple modalities.

Consider a movie scene represented by video and audio tokens. While these tokens naturally align in time, each one can be associated with any other tokens presented in the scene, as shown in Figure 1(a). While the Self-Attention module is effective for computing local associations between tokens, we have observed several drawbacks in the current attention mechanism, especially when tokens originate from diverse modalities. Firstly, different modalities introduce varying granularities of information, leading to potential challenges. Each token in one modality may be associated with multiple tokens in the other modality. Such associations can extend beyond one-to-one correspondences, forming between sub-sequences of tokens in each modality. In Figure 1(a), "Joanna's shouts" might not be associated with a single video token but with several. Moreover, while longer sequences of tokens generally offer richer information, the computational demands of attention mechanisms increase with the input length of tokens. This constraint hinders the effective processing of a higher number of tokens.

Our method stems from the empirical observation that not all tokens in complex input sequences carry equal importance. While prior works like Fan et al. (2021); Tang et al. (2021); Lin & Joe (2023); Rende et al. (2024) have demonstrated the effectiveness of dynamically updated masking mechanisms, this concept remains relatively unexplored in the computer vision domain, with only a few studies such as Lin et al. (2022) exploring it. This gap in vision research has motivated our comprehensive analysis of the impact of dynamic token masking across diverse vision tasks.

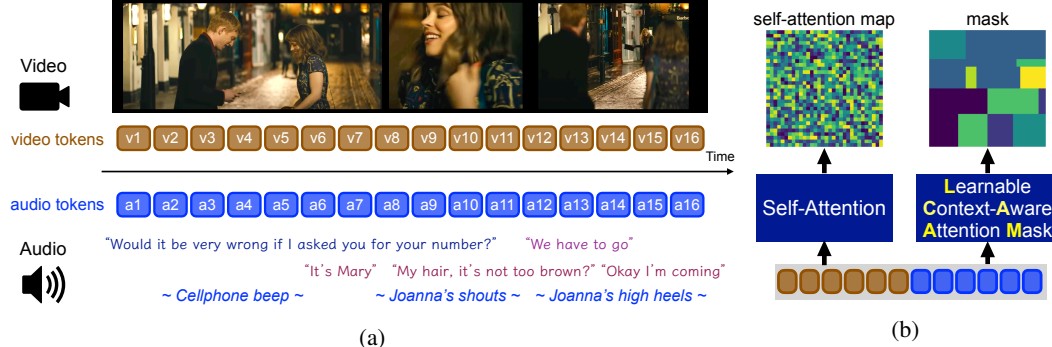

(a)                    (b)

Figure 1: (a) While video and audio tokens naturally align in time, their associations can extend beyond temporal boundaries. For example, "Joanna's shouts" may correspond to multiple video tokens (i.e. not just v8-11, but also v13-16). (b) The Self-Attention module Vaswani et al. (2017) can capture these attention scores *locally*, token-versus-token. We introduce the **Learnable Context-Aware Attention Mask (LCAAM)**, a novel concept that enables a holistic overview of the entire sequence of input tokens, generating a mask that captures attention structures *globally*.

To tackle the challenges posed by complex input sequences, we propose the **Learnable Context-Aware Attention Mask (LCAAM)**[1]—a mechanism that dynamically generates a matrix of weights to regulate attention maps and prioritize critical tokens based on their contextual significance. LCAAM takes as input a token sequence and outputs a mask, capturing attention structures globally across the sequence. Here, "context" encompasses several dimensions: temporal relationships, intermodal associations, and dynamic changes throughout the input sequence. By processing the entire input sequence, LCAAM enables efficient token inspection and dynamic prioritization tailored to each sequence. This adaptive masking technique integrates into existing Transformer Encoder architectures, offering a flexible solution to enhance transformer-based models across diverse applications.

More formally, the LCAAM method employs linear layers that take as input a sequence of tokens with shape $(B, N, E)$, where $B$ is the batch size, $N$ is the number of tokens, and $E$ is the embedding size. The sequence can be either single-modal or multimodal. The output is a mask of size $(B, N, N)$, where for each item in the batch, a mask is generated to capture the attention between all pairs of tokens in the sequence. This generated mask can be applied either globally across all transformer layers in the stack or scaled individually for each transformer layer.

Eventually, the resulting mask, as illustrated in Figure 1(b), captures attention structures globally. This generated mask is then element-wise multiplied or added (depending on the setup) with the attention scores, allowing the masking out or prioritization of specific tokens. Furthermore, our observation that each layer in the Transformer Network embeds different information aspects motivates us to install the LCAAM module per stage, leading to the extension of LCAAM to a multi-layer configuration.

We validate the effectiveness of our approach across various experimental settings. Initially, we assess the capability of the multi-layer LCAAM in multimodal settings, presenting results from audio description generation experiments on the MADv2 dataset (Soldan et al., 2022; Han et al., 2023b). Additionally, we apply our approach to Moment Retrieval and Highlight Detection tasks on the QVHighlight dataset (Lei et al., 2021), incorporating both text and video inputs. Furthermore, we demonstrate that LCAAM can enhance performance in single-modality settings, such as image classification tasks in ImageNet 1K (Deng et al., 2009) and video captioning tasks in MSRVTT (Xu et al., 2016), where a single modality is considered as input to the model. While the performance gain in single-modality settings is modest, we demonstrate that our multi-layer LCAAM can be adopted in various scenarios. Finally, we analyze how the generated mask effectively regulates attention maps.

---

[1]While the term "mask" is used, it is important to note that LCAAM operates as a soft mask or filter, adjusting attention scores rather than fully masking tokens. This allows for more effective prioritization of important information in multimodal sequences.

In summary, our contributions are **three-fold**:

1. We propose the Learnable Context-Aware Attention Mask (LCAAM), a mechanism that dynamically generates masks to regulate attention maps and prioritize significant tokens based on their contextual significance. LCAAM can be integrated into existing Transformer Encoder architectures, potentially enhancing performance on complex sequence processing tasks with minimal architectural modifications.

2. Through experiments across various benchmarks, including MAD, ImageNet-1K, MSRVTT, and QVHighlights, we empirically demonstrate the effectiveness of our LCAAM method, particularly when employed with multimodal encoders.

3. We analyze the output of LCAAM and its influence on attention weight distributions, supplemented by qualitative analysis to provide insights into its behavior.

## 2 RELATED WORK

### 2.1 MULTIMODAL TRANSFORMERS

A predominant area of prior exploration in aligning multiple modalities centers around contrastive learning, a method extensively utilized in both image-text and video-audio alignment contexts (Chen et al., 2020; Khosla et al., 2020; Radford et al., 2021; He et al., 2019; Han et al., 2023a; Zhang et al., 2023a). Recent investigations have also delved into merging diverse modalities within a unified feature space through the incorporation of cross-attention layers (Chen et al., 2021; Lee et al., 2021; Wei et al., 2020; Moon et al., 2023). Furthermore, there is a growing trend of leveraging Transformer capabilities for multimodal fusion tasks (Luo et al., 2021; Kamath et al., 2021; Han et al., 2023a; Barrios et al., 2023; Lei et al., 2021). Our decision to employ a multimodal transformer in our design is rooted in its unparalleled capability to integrate information across diverse modalities, thus fostering a more comprehensive understanding of the input data. Through the utilization of this unified architecture, we are enabled to effectively capture intricate interactions within the sequence, strategically prioritizing relevant cues based on their significance. In contrast to conventional methodologies that treat modalities in isolation, the multimodal transformer facilitates the seamless integration of contextual information, thereby yielding more coherent and nuanced representations.

### 2.2 LANGUAGE MODELS FOR VIDEO DESCRIPTION

To adapt a Large Language Model (LLM) for AD generation, we incorporate an adapter module. This module processes audiovisual features and transforms them into the feature space of our LLM. The concept of training an adapter module rather than finetuning the entire LLM to account for a new modality has been widely explored (Yi-Lin Sung, 2022; Hu et al., 2023), but the method most similar to ours is LLaMA-Adapter (Zhang et al., 2023b; Gao et al., 2023). LLaMA adapter, however, does not account for audio data. Our method follows that of LLaMA-Adapter closely, but changes the input feature space to include both audio and video features. The previous State-of-the-Art in our specific task (generating audio descriptions of movie clips) on the MAD dataset are the AutoAD models Han et al. (2023b;c). We are able to generate comparable results with significantly less fine tuning and contextual information. Recent models have also achieved significant results in finding important moments in longer videos, but these contributions are not particularly relevant to ours because we focus on describing shorter video segments (Lei et al., 2021; Barrios et al., 2023). Another recent result similar to ours is the Video-LLaMA model, which focuses on general purpose visual question answering but uses a Q-Former instead of an adapter module to fuse the visual, audio, and text modalities (Zhang et al., 2023a).

### 2.3 MASKING ATTENTION

In the field of Natural Language Processing, researchers have explored various methods of constructing attention masks, while also investigating their impact on transformer architectures (Fan et al., 2021; Tang et al., 2021; Lin & Joe, 2023; Rende et al., 2024). Conversely, this exploration has received limited attention in Computer Vision (Li et al., 2021; Lin et al., 2022). Motivated by this disparity, our objective is to investigate this phenomenon, particularly in the context of multimodal

data, and its implications for task performance. Unlike the approach proposed by SwinBert (Lin et al., 2022), which advocates for a sparse and learnable mask, our focus aligns more closely with the principles of Mask Attention Networks (Fan et al., 2021). Instead of relying on a static mask matrix, which may restrict the model's ability to capture local relationships effectively, we propose employing a Learnable Context-Award Attention Mask (LCAAM). This adaptive mechanism aims to prioritize and regulate attention tokens within long sequences based on their contextual significance in a dynamic manner.

## 3 LEARNABLE CONTEXT-AWARE ATTENTION MASK (LCAAM)

Our goal is to train a Learnable Context-Aware Attention Mask that effectively prioritizes and regulates tokens based on their significance within a complex sequence. The term "context" here encompasses multiple dimensions: temporal relationships, intermodal associations and dynamic changes throughout the input sequence. This adaptable mechanism can also be seamlessly incorporated into any of the existing Transformer Encoders. Figure 2 shows the overview of the LCAAM architecture.

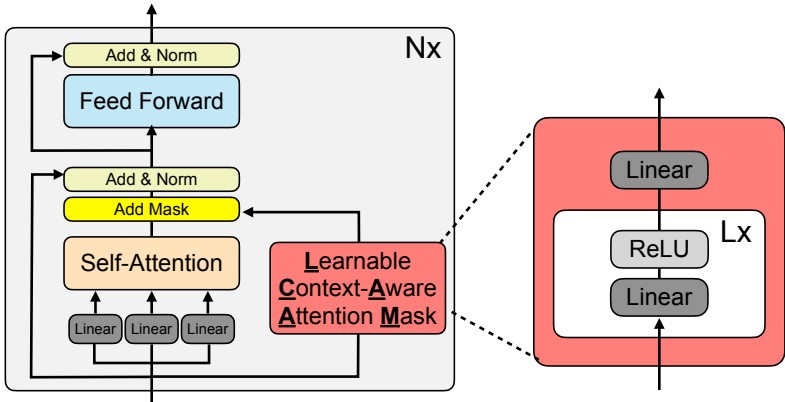

Figure 2: Overview of the Learnable Context-Aware Attention Mask (LCAAM). The LCAAM module processes the entire input sequence to generate a mask. This mask is applied element-wise (e.g., via addition or multiplication, as detailed in Section 3.3) to modify the attention scores produced by the Transformer Encoders.

### 3.1 DEFINITION

We aim to generate an attention mask that adeptly prioritizes and regulates tokens according to their importance within a sequence. For this purpose, we develop a learnable module (denoted as $\mathcal{F}$), which receives a sequence of tokens $\mathbf{X}$ as input and returns a mask $\mathbf{M}$, as shown in Equation 1. The shape of the mask $\mathbf{M}$ depends on the sequence length and the purpose of the multi-head attention, whether it is self-attention or cross-attention.

$$\mathcal{F}(\mathbf{X}) \to \mathbf{M} \tag{1}$$

**Attention.** In the context of self-attention, the resulting mask output can be represented as $(B, N, N)$, where $B$ is the batch size and $N$ is the number of tokens. For instance, consider a multimodal sequence with 128 tokens and a batch size of 1. In this case, the output of our LCAAM would have the dimensions $(1, 128, 128)$. In the cross-attention setup, the mask shape is given by $(B, N_q, N_k)$, where $N_q$ represents the length of the Query tensor and $N_k$ represents the length of the Key tensor. For example, if we perform cross-attention with 75 visual features as the query, 32 text features as the key, and a batch size of 1, the output of LCAAM will be $(1, 75, 32)$. To ensure clarity for the reader, all content in the Section 3 refers to the self-attention scenario unless otherwise specified.

**Scalability.** The mask can be applied either globally across all transformer layers in the stack (Section 3.2) or individually for each layer (Section 3.4). This flexibility allows for different masks to be used at various depths, meaning that each layer can have its own set of learnable parameters, capturing hierarchical information at different levels.

## 3.2 LEARNABLE CONTEXT-AWARE ATTENTION MASK MODULE

The Learnable Context-Aware Attention Mask (LCAAM) module processes an input sequence through a stack of linear layers with ReLU activations. Let $\mathbf{X} \in \mathbb{R}^{B \times N \times D}$ denote the input to the LCAAM module, where $B$ is the batch size, $N$ is the number of tokens, and $D$ is the embedding dimension. The module consists of $L$ layers, where each layer $i \in 1, 2, \ldots, L$ is defined by a weight matrix $\mathbf{W}_i \in \mathbb{R}^{D\mathrm{in}_i \times D\mathrm{out}_i}$ and a bias vector $\mathbf{b}_i \in \mathbb{R}^{D\mathrm{out}_i}$. The forward pass of the LCAAM module can be described as follows:

$$\mathbf{H}_1 = \mathrm{ReLU}(\mathbf{X}\mathbf{W}_1 + \mathbf{b}_1) \tag{2}$$

$$\mathbf{H}_i = \mathrm{ReLU}(\mathbf{H}_{i-1}\mathbf{W}_i + \mathbf{b}_i), \text{ for } i \in \{2, 3, \ldots, L-1\} \tag{3}$$

$$\mathbf{M} = \mathbf{H}_{L-1}\mathbf{W}_L + \mathbf{b}_L \tag{4}$$

Where $\mathrm{ReLU}(\cdot)$ represents the rectified linear unit activation function, $\mathbf{H}_i \in \mathbb{R}^{B \times N \times D_{\mathrm{out}_i}}$ is the output of the $i$-th layer, and $\mathbf{M} \in \mathbb{R}^{B \times N \times N}$ is the generated mask. Note that the input $\mathbf{X}$ and all intermediate outputs $\mathbf{H}_i$ maintain the batch and number of tokens $(B, N)$. For optimal implementation, LCAAM's dimensionality should align with the Transformer Layer.

## 3.3 LEARNABLE CONTEXT-AWARE ATTENTION MASK FUSION

We implement a Learnable Context-Aware Attention Mask for any type of Transformer Layer (Vaswani et al., 2017). For each Transformer Layer, with input $\mathbf{X}$, and learnable mask $\mathbf{M}_i$, the attention mechanism is expressed as:

$$\mathrm{Attention} = \mathrm{softmax}\left(\frac{QK^\top}{\sqrt{d_k}} \diamond \mathbf{M}\right) \tag{5}$$

Here, $Q$ and $K$ are query and key projections of $X$, $d_k$ is the key dimension, and $\diamond$ represents the fusion operation. This fusion can be implemented in two ways: addition ($A \diamond B = A + B$) or element-wise multiplication ($A \diamond B = A \odot B$). The element-wise multiplication, denoted by $\odot$, applies the operation to corresponding elements of the matrices. We tested both methods in our Ablation Studies (Section A.4.1). By default, we use element-wise multiplication in Section 4. When a different operation is employed, we specify it explicitly.

## 3.4 MULTI-LAYER LEARNABLE CONTEXT-AWARE ATTENTION MASK FUSION

We extend the Learnable Context-Aware Attention Mask (LCAAM) approach to a stack of $L$ Transformer layers, with each layer $l$ having its own unique learnable mask, denoted as $\mathbf{M}_l$. The attention mechanism at each layer is computed according to Equation 5. The masks $\mathbf{M}_l$ are learned independently for each layer, allowing each LCAAM to adapt its masking operation based on the layer's representation. As the output of one layer serves as the input to the next, this allows for hierarchical representation learning across the stack.

## 4 EXPERIMENTS

### 4.1 DATASETS

**Generating AD.** MADv2 (Soldan et al., 2022; Han et al., 2023b) is a vast dataset for video-language grounding, with over 264K queries in 488 movies totaling 892 hours. It includes MADv2-eval, with 10 movies for evaluation.

**Moment Retrieval and Highlights Detection.** QVHighlights (Lei et al., 2021) is the latest dataset for moment retrieval and highlight detection, featuring annotations for both tasks in over 10,000 YouTube videos.

**Image Classification.** ImageNet 1K (Deng et al., 2009) is a benchmark dataset consisting of 1.2 million images across 1,000 categories, commonly used for image classification tasks.

**Video Captioning Task.** MSRVTT (Xu et al., 2016) is a dataset for video captioning, comprising 10,000 video clips from 20 categories with human-annotated descriptions.

## 4.2 METRICS

**Generating AD.** Conventional metrics like Rouge-L (R-L)(Lin, 2004), CIDEr (C)(Vedantam et al., 2014), and Retrieval-based metric (R@k/N) (Han et al., 2023c) are employed to compare generated Audio Descriptions (AD) with ground-truth AD. These metrics are robust to low-level variations in testing data, with higher values indicating superior text generation.

**Moment Retrieval and Highlights Detection.** For video grounding tasks, evaluation metrics include Recall@$K$ and mAP@$K$ for IoU=$\theta$ (R@$K$-IoU=$\theta$), assessing both ranking and temporal overlap. Models are evaluated at $K = 1$ with IoU thresholds of 0.5 and 0.7. Average mAP across IoU thresholds from 0.5 to 0.95 with 0.05 increments is calculated. Highlight detection primarily employs mAP, while HIT@1 measures the hit ratio for the highest scored clip.

**Video Captioning.** Evaluation metrics for video captioning include BLEU4 (B4) (Papineni et al., 2002), CIDEr (C), SPICE (S) (Anderson et al., 2016), METEOR (M) (Lavie & Agarwal, 2007), and Rouge-L (R-L), capturing different aspects of caption quality such as n-gram overlap, semantic similarity, and linguistic fluency.

**Image Classification.** Performance in image classification is often measured using Accuracy top-1 (Acc-top1) and Accuracy top-5 (Acc-top5).

## 4.3 BASELINES

The proposed LCAAM module can be incorporated into any of the existing Transformer Encoders. We integrated our contribution into four baseline models: LlaMA AdapterV2 (Gao et al., 2023) with a transformer-based audiovisual encoder, QD-DETR (Moon et al., 2023), SwinBERT (Lin et al., 2022), and ViT Base (He et al., 2021; Vaswani et al., 2017). Our module, described in Section 3, was added **only** to the encoder of each model, except for SwinBERT, where we replaced its fixed learnable mask with our approach to align with its design principle. For more details, see supplementary material.

Table 1: **Comparing performance across various datasets.** We evaluate our masking method on both multimodal encoders and single modality encoders. Our method demonstrates significant performance gains when applied to multimodal encoders, particularly in tasks (a, b, and c). However, for tasks involving single-modality encoders (d and e), we observe minimal improvements across most metrics. The asterisk (*) denotes that we retrained using the codebase and observed a slight decrease in performance compared to the numbers reported in He et al. (2021).

| Model | R-L | C | R@5/16 |
|---|---|---|---|
| LlaMA Adapter (Gao et al., 2023) | 10.7 | 9.4 | 43.4 |
| **Ours** | **13.5** | **18.6** | **56.1** |
| **Gain($\triangle$)** | 2.8 | 9.2 | 12.7 |

(a) AD Task on MADv2-named (Soldan et al., 2022; Han et al., 2023b)

| Model | R1@IoU0.7 | mAP (Avg) |
|---|---|---|
| QD-DETR (Moon et al., 2023) | 44.98 | 39.86 |
| **Ours** | **46.94** | **42.32** |
| **Gain($\triangle$)** | 1.96 | 2.46 |

(b) Moment Retrieval Task in QVHighlights (Lei et al., 2021)

| Model | mAP | HIT@1 |
|---|---|---|
| QD-DETR (Moon et al., 2023) | 38.94 | 62.40 |
| **Ours** | **39.70** | **63.33** |
| **Gain($\triangle$)** | 0.76 | 0.93 |

(c) Highlights Detection at VeryGood confidence in QVHighlights (Lei et al., 2021)

| Model | Acc-Top1 | Acc-Top5 |
|---|---|---|
| *ViT Base (He et al., 2021) | 82.71 | 96.32 |
| **Ours** | **83.45** | **96.59** |
| **Gain($\triangle$)** | 0.74 | 0.27 |

(d) Image Classification in ImageNet 1K (Deng et al., 2009)

| Model | B4 | R-L | M | C | S |
|---|---|---|---|---|---|
| SwinBERT (Lin et al., 2022) | **42.82** | **62.06** | 30.39 | 51.96 | 7.64 |
| **Ours** | 42.03 | 62.05 | **30.60** | **52.24** | **8.03** |
| **Gain($\triangle$)** | −0.79 | −0.01 | 0.21 | 0.28 | 0.39 |

(e) Video Captioning Task in MSRVTT (Xu et al., 2016)

## 4.4 RESULTS

Our comprehensive evaluation across five diverse tasks, as presented in Table 1, reveals the significant impact of our proposed method. The results demonstrate a marked improvement in performance when applied to multimodal encoders, in contrast to single-modality encoders. Specifically, for multimodal tasks, we observe substantial gains: Table 1a shows a maximum improvement of 12.7 points for the R@5/16 metric, with an average improvement of 8.23 points across all metrics. This trend is corroborated by the results in Tables 1b and 1c, where we note maximum improvements of 2.46 and 0.93 points, accompanied by average improvements of 2.21 and 0.86 points, respectively. In contrast, the application of our method to single-modality encoders yields minimal gains and, in some cases, marginal performance decreases. For instance, Table 1e indicates a slight decrease of 0.79 points for the B4 metric, while modest increases of 0.28 and 0.39 points are observed for the C and S metrics, respectively. Similarly, Table 1d shows a nominal improvement of 0.74 points for the Acc-Top1 metric.

**Takeaway:** Our method demonstrates an enhanced capacity to leverage multimodal sequences compared to single-modality inputs. While we observe modest gains in some single-modality tasks, the more pronounced improvements in multimodal scenarios suggest greater potential for advancing multimodal learning tasks. These results indicate promising directions for future research, particularly in optimizing our approach for both multimodal and single-modality applications, with plenty of room for further enhancements across diverse task domains.

## 4.5 ABLATION STUDIES

In this section, we present a comprehensive series of ablation studies to rigorously evaluate the effectiveness of our Learnable Context-Aware Attention Mask (LCAAM) architecture across multiple dimensions. These studies isolate and quantify the contributions of LCAAM in improving model performance, while providing insights into its computational efficiency and scalability. We systematically explore the impact of various attention mask architectures, assess the influence of LCAAM versus parameter scaling, and measure the computational trade-offs associated with its implementation.

Additionally, we analyze how LCAAM scales with increasingly complex multimodal datasets, demonstrating its robustness across varied data sizes and modalities. Further investigations into fusion strategies for masking (e.g., element-wise addition vs. multiplication) and the effects of LCAAM depth, along with detailed visualizations, are included in the Supplementary Material. Our analysis highlights that the LCAAM module offers superior performance gains with minimal computational overhead, positioning it as an effective and scalable solution for modern AI tasks. This evidence supports its broader adoption in multimodal learning and enhances our understanding of optimized attention mechanisms for high-dimensional and dynamic datasets, such as MADv2 and QVHighlights.

Table 2: **Attention Mask Influence.** We analyze the performance of the Audio Description generation task using a subset of 1,010 instances from the MADv2 dataset (see Supplementary Material). Our results show that integrating the Learnable Context-Aware Attention Module (LCAAM) improves performance. Additionally, a more noticeable improvement is observed when using a multi-layer setup, where each transformer attention layer utilizes its own LCAAM, rather than a global one, as seen in row 3. The '*' indicates that we followed the learnable sparse mask design from Lin et al. (2022). All experiments were trained using identical hyperparameters over 10 epochs.

| Mask | R-L | C |
|---|---|---|
| Full Attention | 12.92 | 15.46 |
| Sparse Learnable Mask* | 10.02 | 9.72 |
| LCAAM | 13.10 | 16.58 |
| Multi-Layer LCAAM | **14.28** | **17.11** |

**Impact of Attention Mask Architectures.** We conducted a comprehensive analysis of the performance impact of various attention mask architectures, focusing on our novel Learnable Context-Aware Attention Mask (LCAAM) module. Evaluations were conducted on a specific subset of the

MADv2 dataset (See Supplementary Material for more details), with performance quantified using the Rouge-L and CIDEr metrics (Table 2). Our investigation encompassed four distinct configurations: (i) full attention as a baseline, (ii) the learnable sparse mask from SwinBert, (iii) our proposed LCAAM, and (iv) an extended Multi-Layer LCAAM. Results demonstrate that the introduction of LCAAM yields a substantial performance improvement over the full attention baseline, with CIDEr scores increasing from 15.46 to 16.58. Notably, the Multi-Layer LCAAM architecture achieved superior performance, reaching a CIDEr score of 17.11. In contrast, SwinBERT's sparse learnable mask exhibited a marked decrease in performance, with CIDEr dropping from 15.46 to 9.72. We attribute this decline to the mask's inability to effectively capture the dynamic nature of MAD-v2 sequences, which are characterized by frequent shot changes, transitions, and complex audio-visual interactions. Our LCAAM architecture shows improved performance in handling these multimodal sequences, indicating its effectiveness for the MAD-v2 video captioning task.

Table 3: **Parameters vs. LCAAM Module Influence.** We present the performance results of three models: the baseline model without masking, a Full Attention Transformer with an equivalent number of parameters to the Multi-Layer Learnable Context-Aware Attetion Mask, and the Multi-Layer LCAAM itself. Generally, our findings suggest that the number of parameters does not directly correlate with performance gains.

| Experiment | R-L | C |
|---|---|---|
| Baseline | 12.92 | 15.46 |
| Full Attention w/ same number params. | 11.23 | 12.87 |
| Multi-Layer LCAAM | **14.28** | **17.11** |

**Performance: Parameters vs. LCAAM Module Influence.** We explore whether the performance gains attributed to our Learnable Context-Aware Attention Mask (LCAAM) module are genuinely due to the module itself or simply the result of an increased number of model parameters. To investigate this, we compare a baseline model with full attention to a multi-layer LCAAM model and another variant with full attention but augmented with additional linear layers to equate the LCAAM model's parameter count. Results from the Audio Description generation task (Table 3) indicate that increasing parameters alone, without LCAAM, leads to a performance decline from 15.46 to 12.87 CIDEr, highlighting that more parameters facilitate over-fitting rather than improving the model's learning capability. Thus, the effectiveness of LCAAM arises from its targeted approach to enhancing token attention, rather than merely adding more parameters.

Table 4: **Computational metrics for AD Generation in MADv2**. The table compares models based on performance (R-L, C), computational cost (FLOPs, MACs), parameters, and latency. The baseline shows moderate performance with 6.930B parameters. The **Same Parameters Exp.** increases resource usage but slightly underperforms. **Multi-Layer LCAAM** achieves the best scores (R-L: 14.28, C: 17.11) with only a minor increase in computational cost, highlighting its efficiency in utilizing resources for improved performance. An upward arrow (↑) indicates that higher values are better, while a downward arrow (↓) indicates that lower values are preferred.

| Model | R-L(↑) | C(↑) | FLOPs(↓) | MACs(↓) | Params(↓) | Latency(↓) |
|---|---|---|---|---|---|---|
| Baseline | 12.92 | 15.46 | **3.39 TFLOPs** | **1.69 TMACs** | **6.930 B** | **87.57 ms** |
| Same number params. | 11.23 | 12.87 | 3.45 TFLOPs | 1.80 TMACs | 7.072 B | 89.12 ms |
| Multi-Layer LCAAM | **14.28** | **17.11** | 3.43 TFLOPs | 1.71 TMACs | 7.072 B | 88.60 ms |

**Computational Overhead.** To address potential concerns regarding the computational complexity introduced by our additional module, we conducted a comprehensive analysis of the models' efficiency. Building upon our previous experiments (Table 3), we evaluated the computational overhead of each model by measuring their floating-point operations (FLOPs) and multiply-accumulate operations (MACs). Table 4 presents these metrics alongside the parameter counts and latency measurements for the baseline, same params exp., and our proposed Multi-Layer LCAAM models. Our findings reveal that the Multi-Layer LCAAM architecture offers a compelling trade-off between computational complexity and performance. While it incurs a modest increase in FLOPs, MACs, and parameters compared to the baseline, it demonstrates superior efficiency relative to the equi-parametric experimental model. Notably, our Multi-Layer LCAAM achieves lower FLOPs and

MACs while maintaining parity in parameter count with the experimental model, resulting in an intermediate latency profile. These results suggest that the LCAAM effectively optimizes the model architecture, enhancing resource utilization without significantly compromising speed or introducing undue complexity.

Table 5: **Maximum Performance Gain by Dataset Size and Modality.** This table, adapted from Table 1, presents the maximum performance gains for each dataset, organized by size and modality (V: Vision, T: Text, A: Audio). The results show that larger datasets with multiple modalities, such as MADv2, typically achieve higher performance gains. Conversely, the smaller MSRVTT dataset demonstrates the lowest performance gain. The main goal is to highlight the largest gaps in performance gains. It is important to note that averaging the metrics can be misleading, as different scales among metrics can influence each other when combined, potentially obscuring the true extent of the gaps.

| Dataset | Size | Modality | Max. Gain |
|---|---|---|---|
| MSRVTT | 6.3 GB | V + T | 0.39 |
| Imagenet 1k | 164 GB | V | 0.74 |
| QVHighlights | $\sim$ 180 GB | V + T | 2.46 |
| MADv2 | $\sim$ 3 TB | V + A + T | 12.70 |

**Dataset Scaling.** It is acknowledged that increasing the number of parameters may influence the scaling behavior of data across various modalities. Table 5, which expands upon Table 1, presents a comprehensive analysis of performance gains across datasets of varying sizes and modalities, encompassing Visual, Textual, and Audio data. Our analysis focuses on the maximum performance gain for each metric, a strategic approach that accentuates significant improvements while mitigating distortions inherent in averaging metrics of disparate magnitudes. This method provides a condensed yet insightful representation of our findings. The results reveal a clear correlation between dataset complexity and the efficacy of our LCAAM approach. Notably, large-scale multimodal datasets, exemplified by MAD ($\sim$ 3 TB, V+A+T), demonstrate the most substantial performance improvements, particularly when employing multi-layer LCAAM. For instance, LCAAM achieves a remarkable gain of 12.7 on the MADv2 dataset. In contrast, smaller, less diverse datasets such as MSRVTT (6.3 GB, V+T) exhibit more modest gains (0.39), while medium-sized uni-modal datasets like ImageNet-1k (164 GB, V) show intermediate improvements (0.74). These findings underscore LCAAM's particular effectiveness in handling complex, high-dimensional data structures. The observed performance gradient across dataset sizes and modalities suggests that LCAAM's ability to capture and adapt to intricate cross-modal relationships scales with data complexity. This scalability justifies the computational cost associated with LCAAM, especially in the context of large-scale models and datasets where traditional fine-tuning approaches may prove insufficient or impractical.

**More Ablation Studies**. Section A.4 analyzes the impact of depth on LCAAM performance, explores two element-wise operations for masking fusion, and discusses how LCAAM modifies attention weight distribution.

**Qualitative Analysis.** We conducted two qualitative analyses on the Audio Description Task: one on temporally aligned video and audio streams, and another on misaligned streams. See Section A.4.3.

## 5    Conclusions and Limitations

The Learnable Context-Aware Attention Mask (LCAAM) significantly enhances attention mechanisms in Transformer models, especially for multimodal tasks. By dynamically prioritizing important tokens, LCAAM boosts model performance in diverse tasks. However, its gains in single-modality scenarios are limited. While LCAAM excels at capturing contextual relationships in multimodal tasks, the improvements in single-modality tasks are modest due to the simpler nature of processing a single input type, which underutilizes LCAAM's dynamic masking. Future work could focus on amplifying LCAAM's impact in single-modality tasks, possibly by incorporating additional contextual information or hybrid strategies combining insights from multiple modalities.

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

# A  APPENDIX

## A.1  MULTIMODAL ENCODER TASKS

We evaluated the effectiveness of the Learnable Context-Aware Attention Mask (LCAAM) across three significant multimodal tasks: Audio Description (AD) Generation, Moment Retrieval, and Highlights Detection. In this section, we first outline the definitions of these tasks (Sections A.1.1 and A.1.2) and show the implementation details of our proposed LCAAM module to each task (Sections A.1.3 and A.1.4).

### A.1.1  AUDIO DESCRIPTION GENERATION

Our task involves adapting a Large Language Model (LLM) to generate Audio Descriptions (AD) in text for a long-form movie $\mathcal{L}$ segmented into short clips $\{c_1, c_2, \ldots, c_N\}$. Each clip encompasses $\mathcal{S}$ samples in the visual stream (represented as $V$) and $S$ samples in the audio stream (denoted as $A$)[2]. Specifically, our goal is to create a text $t_i$ that describes the audiovisual content presented in each clip $c_i$, aiming to assist individuals who are blind in following the movie's narrative.

**Audiovisual Model $\mathcal{AV}$.** We aim to train an audiovisual model that comprehends the relationships between sequential video and audio streams. Consequently, $\mathcal{AV}$ processes video ($V$) and audio ($A$) observations sampled at $c_i$ clip and produces an audiovisual feature representations $E_{va}$.

$$\mathcal{AV}(V, A) \rightarrow E_{va} \tag{6}$$

**Large Language Model $\mathcal{H}$.** Given an input sequence $X = \{x_1, x_2, \ldots, x_n\}$, the model $\mathcal{H}$ estimates the probability distribution of the next word $x_{n+1}$ based on the context using the chain rule of probability:

$$P(x_{n+1}|X) = P(x_{n+1}|x_1, x_2, ..., x_n) \tag{7}$$

The model is trained by maximizing the likelihood of generating the correct sequence according to the training data. During inference, it predicts the most likely next word given the context. The model's weights $\theta$ are optimized through back-propagation and gradient descent to improve its language understanding and generation capabilities.

---

[2]Raw sound from movies, excluding descriptions

**Adapter Module $\mathcal{P}$.** Let's assume a pre-trained model with parameters represented by $\theta$. The adapter layer introduces additional parameters for audiovisual understanding task, and these parameters can be denoted as $\phi$. The output of the adapter layers can be represented as $P(x', \phi)$, where $x'$ is the projected audiovisual features into the language space. So, the overall output of the model with the adapter layer can be written as:

$$\mathcal{F}(\mathcal{H}(x, \theta), \mathcal{P}(x', \phi)) \to t_i \tag{8}$$

Where $\mathcal{F}$ is a function that combines the pre-trained Language Model $\mathcal{H}$ and the adapter $\mathcal{P}$ to produce an AD in text $t_i$.

### A.1.2 MOMENT RETRIEVAL AND HIGHLIGHTS DETECTION

The visual-language grounding model, denoted as $\mathcal{G}$, is tasked with processing an untrimmed video, $V$, sampled from a temporal window $W$, along with a natural language query $Q$. It then produces predictions for $J$ temporal moments, defined as:

$$\mathcal{G}(V, Q) \to (\tau_s, \tau_e, s, s_l)_1^J. \tag{9}$$

In Equation 9, the grounding models yield a series of moments ranked by their confidence scores. Here, $(\tau_s, \tau_e)$ represents the duration span of the moment, while $s$ indicates its confidence score. Now, let's define the inputs for our attention modules. Given a video comprising $L$ clips and a text query containing $N$ words, their representations extracted by frozen video and text encoders are denoted as $v_1, v_2, \ldots, v_L$ and $t_1, t_2, \ldots, t_N$, respectively. Additionally, the grounding model provides saliency scores $s_l$ for each moment for the highlight detection task.

### A.1.3 IMPLEMENTATION DETAILS FOR AD GENERATION

**Feature Extraction.** The extraction of visual features follows the CLIP-based methodology outlined in Soldan et al. (2022). To be more specific, visual features are extracted at a rate of 5 frames per second (FPS) with an embedding dimensionality of $D_v$=512. For audio feature extraction, we follow Barrios et al. (2023) by utilizing the OpenL3 (Cramer et al., 2019; Arandjelovic & Zisserman, 2017) checkpoint pre-trained on videos containing environmental audiovisual data. We use a spectrogram time-frequency representation with 128 bands and set the audio embedding dimensionality $D_a$ to 512. Furthermore, we extract the audio embeddings using a stride size of 0.2 seconds, *i.e.*, with an extraction frame rate of 5 Hz, matching the frame rate of the visual features.

**Audiovisual Model $\mathcal{AV}$.** We utilize a Multimodal Transformer with a standard configuration (Vaswani et al., 2017). For each observation $c_i$, consisting of both visual and audio information, we employ $S = 25$ visual tokens and $S = 25$ audio tokens, effectively spanning a 5-second duration at a frame rate of 5 FPS. This Multimodal Transformer architecture comprises 16 layers and employs a Multi-Layer Learnable Context-Aware Attention Mask Module with a dimensionality of 768 and depth of 16.

**Large Language Model $\mathcal{H}$.** For Large Language Model, we choose to employ a frozen LLaMA 7B model (Touvron et al., 2023) and opt to use its official checkpoint.

**Adapter Module $\mathcal{P}$.** We build our audiovisual adapter following the approach done in Gao et al. (2023). In this part, we select 16 tokens as audiovisual tokens. We adjust the last 31 layers of LLaMA 7B, making sure that the audiovisual features stay at a size of 512, which then maps to 4096 (LLaMA dimensionality). We set the depth to 8, use 16 heads, apply LoRA Rank (Hu et al., 2021) with a value of 16, and activate Bias layers (Zhang et al., 2023b).

**Training Protocol.** To generate Audio Descriptions, we follow the training methodology outlined in Zhang et al. (2023b); Gao et al. (2023). This involved utilizing 8 RTX 6000 Ada Generation GPUs, each equipped with 50 GB VRAM, alongside employing a base learning rate of $1e-4$ and the Adam optimizer.

### A.1.4 IMPLEMENTATION DETAILS FOR MOMENT RETRIEVAL AND HIGHLIGHTING TASK

**Feature Extraction.** The visual and text embeddings are extracted following the methodology presented in Lei et al. (2021). For video, we use SlowFast (Feichtenhofer et al., 2018) and the visual encoder (ViT-B/32) of CLIP (Radford et al., 2021) to extract features every 2 seconds. We then normalize the two features and concatenate them at hidden dimension. The resulting visual features is denoted as $E_V \in \mathbb{R}^{L_V \times D_V}$, with $D_V = 2816$. For text features , we use the CLIP text encoder to extract token level features, $E_V \in \mathbb{R}^{L_Q \times D_Q}$ with $D_V = 512$.

**Video Grounding Model.** We adopt the methodology outlined in Moon et al. (2023). The architecture consists of three distinct components: an encoder comprising four layers of transformer blocks (two cross-attention layers and two self-attention layers), while the decoder has only two layers. We configure the hidden dimension of the transformers to be 256 Additionally, for the transformer encoder layers and the cross-attention layers, we utilize our LAACM using dimensionality of 256 and depth of 32 layers.

**Training Protocol.** We conducted training over 200 epochs, employing a batch size of 32 and a learning rate set to $1e-4$. We utilized the Adam optimizer with a weight decay of $1e-4$, leveraging a single GPU, the RTX 6000 Ada Generation.

## A.2 SINGLE MODALITY ENCODER TASKS

### A.2.1 IMAGE CLASSIFICATION TASK

In the image classification task, the goal is to assign an input image $I$ to one or more predefined classes from a set of $C$ classes. Let's denote the image classification model as $\mathcal{M}$. Given an input image $I$, the model generates a set of class predictions and their corresponding confidence scores:

$$\mathcal{M}(I) \rightarrow (\hat{y}_1, \hat{p}_1), (\hat{y}_2, \hat{p}_2), \ldots, (\hat{y}_C, \hat{p}_C) \tag{10}$$

Here, $\hat{y}_c \in 1, 2, \ldots, C$ represents the predicted class label for the $c$-th class, and $\hat{p}_c \in [0, 1]$ is the corresponding confidence score or probability assigned by the model to that class. The model's goal is to accurately predict the true classes present in the input image $I$.

### A.2.2 VIDEO CAPTIONING TASK

In the video captioning task, the goal is to generate a textual description or caption for a given input video $V$. Let's denote the video captioning model as $\mathcal{M}$. Given an input video $V$, the model generates a sequence of words $W = w_1, w_2, \ldots, w_N$ that forms the caption:

$$\mathcal{M}(V) \rightarrow W = w_1, w_2, \ldots, w_N \tag{11}$$

Here, each $w_i$ represents a word in the generated caption, and $N$ is the length of the caption sequence. The model's objective is to produce a natural language caption $W$ that accurately and coherently describes the content and events depicted in the input video $V$.

### A.2.3 IMPLEMENTATIONS DETAILS FOR IMAGE CLASSIFICATION TASK

We follow the pre-trained model developed in He et al. (2021) and fine-tune it for the image classification task. The base model is a Vision Transformer (ViT) with a 16x16 patch size, 768-dimensional embedding, 12 transformer layers, and 12 attention heads. It includes an MLP ratio of 4, biases in the query, key, and value projections, and layer normalization with an epsilon of $1e - 6$. To incorporate our proposed Learnable Context-Aware Attention Mask (LCAAM) module, we use the Multi-Layer LCAAM variant, which generates the attention mask using a single linear layer. For the pretraining stage, we adhere to the methodology outlined in He et al. (2021), but increase the batch size to 128 and use 4 gradient accumulation steps. For fine-tuning on the image classification task, we maintain a batch size of 128 and 4 gradient accumulation steps. Additionally, we train for 100 epochs, apply a weight decay of 0.05, set the drop path rate to 0.1, and use mixup and cutmix with values of 0.8 and 1.0, respectively.

### A.2.4 IMPLEMENTATIONS DETAILS FOR VIDEO CAPTIONING TASK

We adopt the methodology proposed by SwinBERT (Lin et al., 2022), with a notable modification. Instead of using a fixed learnable mask implemented via `nn.Parameter`, we integrate our

Learnable Context-Aware Attention Mask (LCAAM) module, which consists of 16 layers while maintaining the same dimensionality as the original SwinBERT. Regarding the hyperparameters, the experiment utilizes a batch size of 2 per GPU, running for 20 epochs with a learning rate of 0.0003. Training is conducted in half precision using DeepSpeed, with gradient accumulation over 16 steps. For the entire training process, we used 8 A6000 Ada generation GPUs.

### A.3 ADDITIONAL DETAILS FOR AUDIO DESCRIPTION GENERATION

In the following sections, we examine specific details that have not been addressed in the main paper. This comprehensive discussion includes insights into the current methodology for calculating metrics, the specific prompts employed, and the intricacies of both the training and evaluation processes for our implementation.

#### A.3.1 METRICS

In this work, we compute the CIDEr (Vedantam et al., 2014) score using the `pycocoeval` package from the coco-caption repository, adhering to the standard parameters of $n = 4$ and $sigma = 6$ as prescribed in Vedantam et al. (2014). For Rouge-L (Lin, 2004), a commonly used metric in natural language processing, we leverage the Hugging Face `evaluate` library for implementation (evaluate-metric/rouge). The Rouge-L configuration is set with `use_aggregator=True` and `use_stemmer=True`, aligning with the default settings to ensure consistent evaluation. Prior to metric computation, both predicted and reference texts are normalized by converting to lowercase and removing punctuation, following standard preprocessing protocols.

For retrieval-based evaluation, we adopt the R@k/N metric, utilizing the methodology introduced in Han et al. (2023c). This is further supplemented by the BERTScore (Zhang et al., 2020) metric, ensuring alignment with state-of-the-art retrieval practices. To maintain reproducibility and result comparability, we use the specified hash code for BERTScore: `roberta-large_L17_noidf_version=0.3.12(hug_trans=4.30.2)-rescaled`, which reflects the model version and Hugging Face environment at the time of evaluation. These standardized configurations and consistent preprocessing steps reinforce the robustness and reliability of our evaluation pipeline.

#### A.3.2 NATURAL LANGUAGE PROMPTING

To implement Audio Description functionality in our model, we apply the prompting approach developed in the LLaMA Adapter framework (Gao et al., 2023). The primary prompt used for generating Audio Descriptions is: **"Below is an instruction that describes a task. Write a response that appropriately completes the request."** We then include a task-specific instruction: **"Generate a caption for this video."** This prompt setup, shown in Figure S3, provides the model with the necessary context to produce relevant and concise descriptions for the video content.

```
Below is an instruction that describes a task
### Instruction:
Generate caption of this video.
### Response:
```

Figure S3: **Prompt for Audio Description Generation** The caption provided outlines the prompt utilized to activate the functionality of Audio Description generation employing the LLaMA model.

#### A.3.3 DATASET SPLIT

As MADv2 lacks a validation set, we curated a subset of 1010 moments from two movies, 3034_IDES_OF_MARCH and 3074_THE_ROOMMATE from the Unnamed version for our ablation studies and model selection. All models and experiments were assessed under consistent parameters to ensure fair comparisons. However, Table 1 in the main paper was generated using the entire dataset in the named version to maintain parity with other baselines.

### A.3.4 Training Protocol

The training procedure for our Audio Description Generation model adhered closely to the methodology outlined in Gao et al. (2023). The process began with an initial alignment phase aimed at ensuring robust synchronization between the audiovisual features. This phase was crucial for establishing coherence between the audio and visual modalities of the input data. Upon successful alignment, we resumed training with a focus on optimizing the bias and gate layers as proposed by Gao et al. (2023), leveraging the LLaMA Touvron et al. (2023) 7B architecture in combination with our audiovisual encoder. In this subsequent stage, we performed backpropagation exclusively on the bias, gate, and audiovisual layers to enhance the model's capacity to generate accurate and contextually relevant audio descriptions.

Training was conducted over a span of 20 epochs, with model selection based on performance on the validation subset. Hyperparameters were meticulously tuned, including a learning rate of $1e^{-4}$, weight decay of 0.05, and a batch size of 256. We employed the AdamW optimizer to ensure efficient parameter updates. During the audiovisual alignment phase, the adapter and audiovisual layers were trained for 2 epochs, with the rest of the model parameters held constant, facilitating stable convergence. Importantly, the LLaMA model's core parameters remained frozen throughout the entire training process, preserving the integrity of its pre-trained features while allowing focused adaptation of the newly introduced layers. This careful balance between alignment and fine-tuning was critical for achieving high-quality audio description generation without disrupting the foundational capabilities of the LLaMA architecture.

### A.4 Ablation Studies

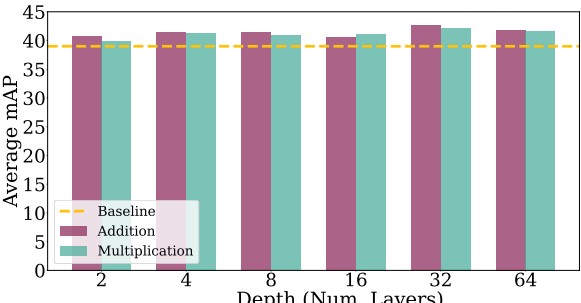

Figure S4: **Ablation Studies on the number of layers in LCAAM and types of mask operation.** We conduct an investigation into the impact of varying the number of layers utilized within the Learnable Context-Aware Attention Mask (LCAAM) framework, as applied in the cross-attention configuration, along with the methods employed for mask fusion with attention weights. The experimentation involves the manipulation of the number of layers, ranging from 2 to 64, and explores two distinct fusion techniques: multiplication and addition operations, both implemented at the element-wise level. Evaluation of these experiments is carried out on the validation split set of QVHighlights (Lei et al., 2021). Overall, notable enhancements in performance, particularly concerning the Average mAP metric for the Moment Retrieval task, are observed. The most substantial improvements are achieved when utilizing 32 layers within the LCAAM module.

### A.4.1 Effects of Depth and Masking Fusion Techniques

Our ablation study on the cross-attention mechanism systematically investigates the impact of two critical components within the Learnable Context-Aware Attention Mask (LCAAM) module: the depth of the LCAAM architecture and the mask operations (addition and multiplication). As illustrated in Figure S4, we evaluate performance using the Average mAP metric for Moment Retrieval on the QVHighlights validation set. The results demonstrate that LCAAM consistently enhances performance across various configurations, with some architectures yielding more substantial improvements than others. Notably, all LCAAM variants, regardless of layer composition or operation type, outperform the baseline model. The most effective configuration utilizes 32 layers with both addition and multiplication operations, achieving Average mAP scores of 42.61 and 42.32, respectively. These findings underscore the efficacy of our approach in bolstering Moment Retrieval performance on the QVHighlights dataset and suggest that the interplay between architectural depth and diverse mask operations is crucial for optimizing cross-attention mechanisms in this context.

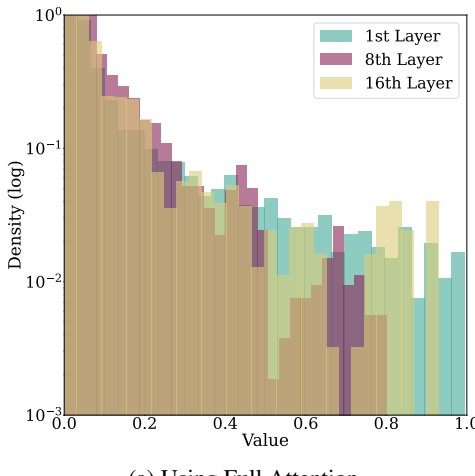 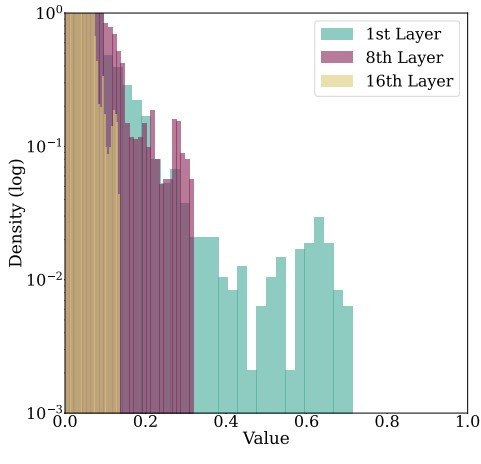

(a) Using Full Attention          (b) Using Multi-Layer LCAAM

Figure S5: **Attention Weight Distribution.** This figure illustrates the effect of our Learnable Context-Aware Attention Mask (LCAAM) on the distribution of attention weights during the AD generation task. When using LCAAM, attention weights tend to decrease in magnitude as they propagate through deeper layers, with many approaching zero. This observation may facilitate future exploration of attention optimization by potentially reducing redundant computations. The attention weights shown were collected from a forward pass using 64 samples.

### A.4.2 ATTENTION WEIGHTS

Figure S5 presents a comparative analysis of attention weight distributions across three critical layers (1st, 8th, and final) of the Transformer architecture, contrasting traditional full-attention mechanisms with our proposed Multi-Layer Learnable Context-Aware Attention Mask (LCAAM). Our findings reveal a striking pattern: the implementation of Multi-Layer LCAAM induces a substantial sparsification of attention weights, with a significant proportion reducing to zero and many others converging to near-zero values. This phenomenon suggests that LCAAM effectively prunes redundant connections within the attention mechanism, potentially leading to more computationally efficient model training without sacrificing performance. The observed sparsity not only aligns with recent trends in neural network optimization but also opens avenues for further research into the interpretability and efficiency of attention-based models. While these results underscore the potential of LCAAM as a promising approach for enhancing the scalability and resource utilization of Transformer-based architectures, there remains considerable room for improvement and further investigation. Future work could explore the optimal degree of sparsity, the impact on various downstream tasks, and potential hybridization with other attention optimization techniques to further push the boundaries of efficient, high-performance Transformer models.

### A.4.3 QUALITATIVE ANALYSIS

Figure S6 presents a qualitative analysis of our Multi Layer Learnable Context-Aware Attention Mask (LCAAM) implementation for the Audio Description generation task. This visualization encompasses two key aspects: Figure S6a displays the concurrent audio and video signals, and Figure S6b illustrates the mask values corresponding to each token in the initial transformer layer. In this figure, the x-axis represents the sequence of tokens, and the colored heatmap indicates the mask values for each token in relation to the other tokens in the sequence. In this example, the first 25 tokens represent visual information, and the last 25 tokens correspond to the audio data. Each token (highlighted in the title) is analyzed in terms of its interaction with the sequence.

In this example, the token sequence has a shape of $(1, 50, 756)$, where 50 denotes the total number of tokens resulting from the concatenation of visual and audio tokens, each contributing 25 tokens. The visual content remains largely static across frames, depicting a residential backyard with minor visual variations. The auditory content transitions from ambient sounds such as wind, insects, and outdoor noise to the rhythmic pattern of a clock. The ground truth Audio Description states: "A

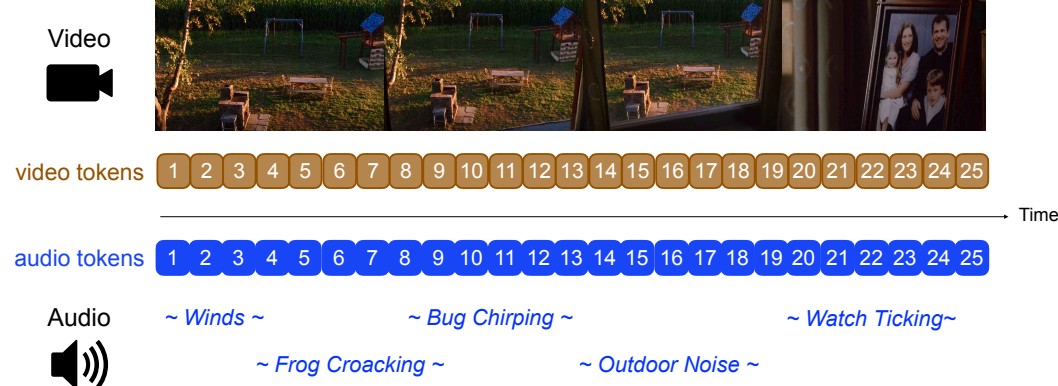

(a) **Scene Visualization.** We highlight a specific moment from the movie Signs (2002) for qualitative analysis within the MADv2-eval set. Here, we meticulously present the visual elements while accurately representing the accompanying audio signals of the scene.

(b) **Scene Visualization** We also showcase the mask values produced by the Learnable Context-Aware Attention Mask (LCAAM) module for each visual and audio token present in the scene. These mask values exhibit positive numerical values, ranging between 0 and 1 inclusively.

Figure S6: **Qualitative Analysis.** This illustration presents a qualitative analysis of a specific instance from the MADv2-eval dataset. It depicts visual and audio signals alongside mask values corresponding to the initial transformer layer (1st layer). Video tokens are represented on the x-axis from 0 to 24, while audio tokens range from 25 to 49 on the same axis. The ground truth label for this moment is: "A set of swings and a climbing frame stand in a rural backyard, along with a picnic table and a brick barbecue."

set of swings and a climbing frame stand in a rural backyard, along with a picnic table and a brick barbecue."

In this scenario, the LCAAM module activates only three out of twenty-five visual tokens while assigning minimal attention to audio tokens. Figure S6b shows the masking values for each token, where the x-axis corresponds to the sequence of tokens resulting from the concatenation: tokens numbered from 0 to 24 are visual tokens, and tokens from 25 to 49 are audio tokens. For instance, for visual token 1, LCAAM assigns values greater than 0.35 to tokens 0 to 24 (the visual tokens), indicating strong correlations between these visual elements, as shown by the yellow-green-colored cells in the heatmap. Moreover, some correlation with audio tokens (25 to 49) is also visible in the

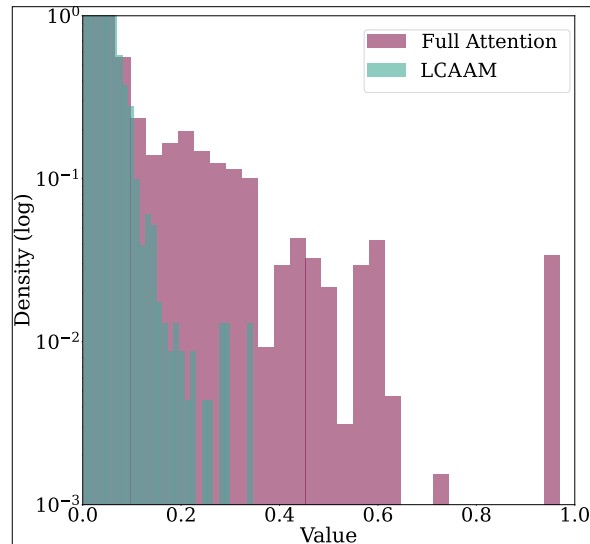

Figure S7: **Analysis of Attention Weight Distribution in the Qualitative Example.** The plot illustrates the distribution of attention weights within the initial transformer layer across two distinct configurations: employing Learnable Context-Aware Attention Mask (LCAAM) and full-attention mechanisms. It is evident from the depiction that attention weights under LAM tend to exhibit a leftward bias, resulting in a significant portion approaching 0 or nearing zero. The distribution weights correspond to the same example in Figure S6.

Figure S8: **Analysis of LCAAM Failure Example in Audio Description Generation.** This plot illustrates the learned mask (LCAAM's output) from the example shown in Figure S6. In this scenario, the visual features remain unchanged, but the audio tokens correspond to the last 25 samples from the movie's credits, which consist solely of the soundtrack. While the mask correctly assigns low values to the visual features, it fails to do so for the audio features, assigning mid-range values from the distribution instead. The x-axis represents the video tokens (ranging from 0 to 24) and the audio tokens (ranging from 25 to 49) on the same axis.

figure, though these values are generally lower. Conversely, for audio token 1, LCAAM assigns higher values to the initial visual tokens and lower values to the later visual tokens, reflecting the static nature of the visual information—a backyard scene with minimal dynamic changes—while the other audio tokens receive varying degrees of attention. Notably, the last audio tokens (e.g Audio Token 15 to 25) correspond to indoor sounds, indicating a scene transition from an outdoor to an indoor setting. Consequently, LCAAM assigns values less than $0.35$ in its masking for these tokens, interpreting them as less important and less related to the predominantly outdoor visual and audio tokens.

To compare with self attention, Figure S7 shows the attention weight distributions for both LCAAM and full attention on the same scene. Without LCAAM, the distribution is more uniform, suggesting

that attention is spread across more tokens. With LCAAM, the distribution is skewed to the left with many weights near zero, implying focused attention on fewer, more relevant tokens. This analysis highlights LCAAM's capability to discern and prioritize specific tokens, thereby enhancing multimodal scene interpretation.

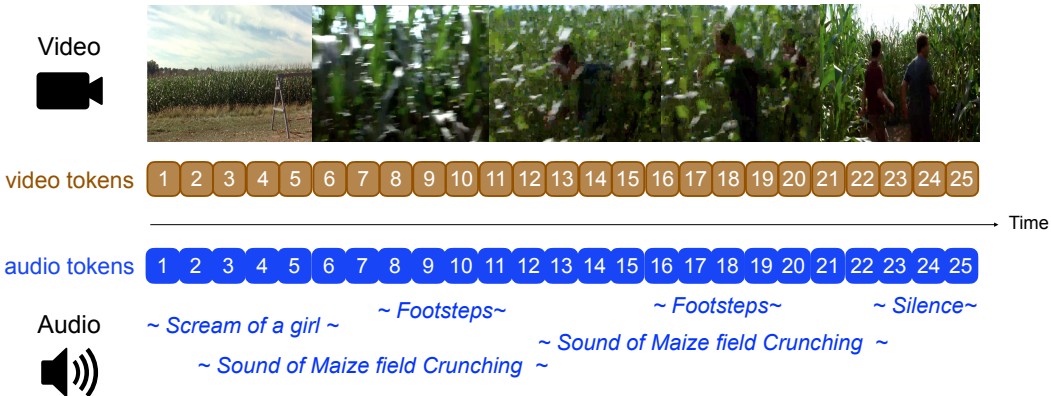

(a) **Scene Visualization.** We highlight a specific moment from the movie Signs (2002) for qualitative analysis within the MADv2-eval set. Here, we meticulously present the visual elements while accurately representing the accompanying audio signals of the scene.

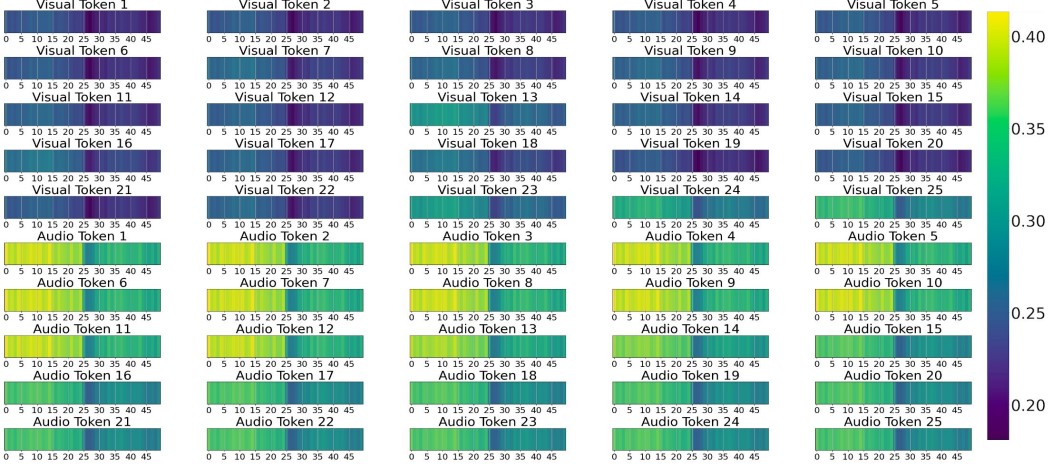

(b) **Scene Visualization** We also showcase the mask values produced by the Learnable Context-Aware Attention Mask (LCAAM) module for each visual and audio token present in the scene. These mask values exhibit positive numerical values, ranging between 0 and 1 inclusively.

Figure S9: **Additional Example for Qualitative Analysis.** This illustration provides an additional example of qualitative analysis from the MADv2-eval dataset. It displays both visual and audio signals along with corresponding mask values from the first transformer layer (1st layer). The x-axis represents video tokens from 0 to 24 and audio tokens from 25 to 49. The ground truth label for this moment is: "They stop when they reach a gap".

Figure S8 illustrates a challenging scenario for our LCAAM approach. This example uses the same visual content as in Figure S6 but pairs it with audio samples comprising 25 tokens from the credits section, containing only background soundtrack music. While LCAAM correctly assigns minimal values to the visual tokens, recognizing the lack of relevance between the video and the new audio tokens, it struggles to handle the audio tokens optimally. Instead of assigning values close to zero to the audio tokens—as would be expected given the irrelevance of the soundtrack to the scene description task—LCAAM assigns intermediate values from its distribution. This outcome suggests

potential areas for improvement in the model's audio-visual integration capabilities, particularly in distinguishing between relevant and irrelevant audio information.

Another example, depicted in Figure S9, involves the ground truth labeled as "They stop when they reach a gap." The scene opens with an image of a maize field, accompanied by the sudden sound of a little girl screaming. The film's protagonists immediately begin sprinting through the field, generating a distinct crunching noise alongside the sound of rapid footsteps. While the visual content is highly dynamic—both the character and the environment are in motion—the scene remains largely focused on the maize field and the actors running through it. This continues until a moment of silence marks their exit from the field. In Figure S9b, the final visual tokens (24 and 25) carry the most weight in the LCAAM output because they show the characters stopping, which aligns with the ground truth. Additionally, the audio of the crunching sound (audio token 1 to 14) from the maize field provides context, as it reflects the running action and comes before the stopping action, which is the task's consequence. The silence that follows signifies the stopping action and the fact that the gap has been crossed, as there is no more maize field to traverse. This is why later audio tokens (20-25) are attended to, though less strongly, as they represent the conclusion of the scene.

In summary, these findings highlight both the strengths and limitations of LCAAM in multimodal Audio Description generation. While the model effectively prioritizes relevant tokens in scenes with aligned audio and visual content, it struggles with irrelevant audio, assigning undue attention to non-informative tokens. This underscores the need for further refinement in its ability to discriminate between pertinent and extraneous information, suggesting avenues for future research to enhance multimodal attention mechanisms. There is still room for improvement, and we are optimistic that addressing these challenges will further advance the effectiveness of LCAAM in complex multimodal tasks.

