# OpenReview forum: "Learnable Context-Aware Attention Mask for Multimodal Transformers"
_ICLR.cc/2025/Conference — Submitted to ICLR 2025_

### Official Review · Reviewer_mkMS · 2024-11-02

**Soundness:** 2
**Presentation:** 2
**Contribution:** 3
**Rating:** 5
**Confidence:** 4

**Summary:**

The paper addresses the issues of varying token granularity and the high computational cost of processing long sequences by proposing a learnable context masking mechanism. This mechanism uses a globally learnable attention map to select the most important tokens for subsequent learning. Extensive experiments are provided to support this approach.

The author's detailed response has addressed most of my questions. Utilizing contextual information to dynamically adjust attention maps to enhance the attention mechanism is highly insightful and valuable. From the experimental results, it is evident that LCAAM effectively prunes redundant connections within the attention mechanism. After carefully considering the reviewer's comments in sa9g, I will appropriately increase my score.

**Strengths:**

1."The paper is logically clear and well-structured."

2."Experiments were conducted on multiple datasets, and additional supplementary materials are provided."

**Weaknesses:**

1."The idea presented in the paper is very interesting, and the problem addressed is important. However, it is not clear from the paper whether the mask mechanism alone can effectively handle token granularity and the high computational cost. The paper does not provide a detailed analysis. Please provide your motivation and explain how the mask tokens address these issues."

2."The paper implements the mask tokens using a few layers of MLP without any additional constraints. Is the output of a few MLP layers directly suitable as mask features? This part lacks a detailed analysis. Why is the output of a few MLP layers effective for the desired result? This approach seems more like an adapter. Please provide a detailed explanation of this aspect."

**Questions:**

1.Is the output of a few MLP layers directly suitable as mask features?
2.Why is the output of a few MLP layers effective for the desired result?

---

> ### Author Response · Authors · 2024-11-22
> **Response to the Reviewer mkMS (Part 1)**
>
> We thank the reviewer for acknowledging the logical structure of the paper and the extensive experiments across multiple datasets. These strengths reflect our commitment to presenting a rigorous and well-substantiated contribution.
>
> ### Weaknesses
>
> 1. **The idea presented in the paper is very interesting, and the problem addressed is important. However, it is not clear from the paper whether the mask mechanism alone can effectively handle token granularity and the high computational cost. The paper does not provide a detailed analysis. Please provide your motivation and explain how the mask tokens address these issues.**
>
> **Section A.4.3** explains how the LCAAM module handles  token granularity. For example, **Figure S6** illustrates that LCAAM activates only three out of twenty-five visual tokens and minimally attends to audio tokens. In **Figure S6b**, tokens 0–24 are visual and 25–49 are audio. Visual token 1 receives high values (>0.35) for visual tokens, indicating strong visual connections (highlighted in yellow-green on the heatmap), while audio tokens receive lower values. Conversely, audio token 1 prioritizes early visual tokens, reflecting the static backyard scene, and assigns lower values to later visual tokens. Additionally, later audio tokens (15–25), which represent indoor sounds, have masking values below 0.35, indicating a transition to an indoor setting.
>
> Another example in **Figure S9** demonstrates the ground truth "They stop when they reach a gap." Here, the final visual tokens (24 and 25) are most significant, aligning with the characters stopping. Audio tokens 1–14 capture running sounds, providing context, while later audio tokens (20–25) indicate the scene’s conclusion with less emphasis.
>
> Regarding computational cost **(discussed in Lines 423–435)**, the Multi-Layer LCAAM architecture offers an excellent balance between complexity and performance. It only slightly increases FLOPs, MACs, and parameters compared to the baseline but is more efficient than the equi-parametric model, using fewer FLOPs and MACs while maintaining the same number of parameters. This optimization enhances resource utilization without significantly affecting speed or adding complexity.
>
> 2. **The paper implements the mask tokens using a few layers of MLP without any additional constraints. Is the output of a few MLP layers directly suitable as mask features? This part lacks a detailed analysis. Why is the output of a few MLP layers effective for the desired result? This approach seems more like an adapter. Please provide a detailed explanation of this aspect.**
>
> To clarify, the LCAAM is implemented using a few layers of MLP followed by a softmax operation. Specifically, the proposed approach applies $M$ **before the softmax operation**, inherently constraining attention values to be non-negative, as softmax produces only positive probabilities. This not only preserves well-defined attention scores but also refines the attention weight distributions, as illustrated in **Figures S5 and S7 (Appendix)**. By prioritizing context-aware token selection, this strategy enhances performance, particularly in tasks like audio description, where nuanced attention is critical.
> The suitability of generated masks is supported by results from Section A.4.1 **(Line 906)**, which show that increasing the depth of the MLP layers improves mAP in the moment retrieval task. This indicates that deeper MLP configurations capture richer contextual relationships and contribute directly to the success of the LCAAM. Furthermore, the analysis of masking fusion techniques in the same section reveals that element-wise multiplication outperforms addition, dynamically scaling attention scores to prioritize tokens effectively. The results confirm that MLP-generated masks are not mere adapters but robust mechanisms for regulating attention and improving performance in token prioritization tasks.

---

> ### Author Response · Authors · 2024-11-22
> **Response to the Reviewer mkMS (Part 2)**
>
> ### Questions
>
> 1. **Is the Output of a Few MLP Layers Directly Suitable as Mask Features?**
>
> Yes, the output of the MLP layers is directly suitable as mask features due to their alignment with the input sequence dimensions and their ability to adapt dynamically to varying attention mechanisms. In Section 3.2, the proposed LCAAM generates masks that correspond precisely to the input tokens, ensuring compatibility for both self-attention and cross-attention setups. For example, as explained in Lines 204–212, a multimodal self-attention sequence with 128 tokens produces a mask of dimensions (1, 128, 128), while a cross-attention setup with 75 visual tokens and 32 text tokens results in a mask of dimensions (1, 75, 32). This adaptability ensures that the masks are task-agnostic and seamlessly integrate with different architectures. Additionally, the results in Section A.4.1 demonstrate that deeper MLP configurations improve performance metrics such as mAP, confirming their effectiveness in generating meaningful, context-aware masks. These features make the MLP-generated masks highly suitable for attention modulation in complex Transformer tasks.
>
> 2. **Why is the Output of a Few MLP Layers Effective for the Desired Result?**
>
> In Section A.4.1, **"Effects of Depth and Masking Fusion Techniques"**, we present an ablation study focused on the cross-attention mechanism within our Learnable Context-Aware Attention Mask (LCAAM) module. This study systematically examines **two critical factors: the depth of the LCAAM architecture and the types of mask** operations employed (addition and multiplication). As illustrated in Figure S4, we evaluated performance using the Average mAP metric for Moment Retrieval on the QVHighlights validation set.
> Our findings demonstrate that incorporating a few MLP layers within the LCAAM module significantly enhances performance across various configurations, consistently outperforming the baseline model. Specifically, a 32-layer LCAAM architecture utilizing both addition and multiplication mask operations achieved the highest Average mAP scores of $42.61$ and $42.32$, respectively. These results highlight that the strategic depth of MLP layers effectively captures complex contextual relationships, thereby optimizing the cross-attention mechanism.

---

> > ### Author Response · Authors · 2024-11-27
> >
> > Dear Reviewer,
> >
> > I hope all your doubts have been resolved. Please don’t hesitate to reach out if you need any further clarification.
> >
> > Kind regards,
> >
> > Authors of paper 4753

---

### Official Review · Reviewer_sa9g · 2024-11-03

**Soundness:** 2
**Presentation:** 1
**Contribution:** 1
**Rating:** 3
**Confidence:** 5

**Summary:**

This paper presents a method that dynamically generates attention score map. The authors claim that in multi-modal settings where inputs of diverse modalities and long sequence lengths make transformers' self attention mechanism hard to perform well. From this motivation, the authors propose an additional approach on top of self-attention that generates attention maps that eventually get incorporated to QK similarity scores by either addition or multiplication.

**Strengths:**

The algorithm is simple enough for readers to follow. The authors present results in multiple benchmarks, achieving compelling accuracies. This paper tackles a problem inherent in multi-modality, which the community has recently shown significant interests.

**Weaknesses:**

I have multiple concerns regarding the paper, as listed below:
- The method part is not explaining the concept correctly. In Sec 3.2, which is supposed to be the core contribution of this paper, the final output of the stack of layers is in shape of $\mathbb{R}^{B \times N \times D_{out_L}}$. Do the authors mean that the output dimension of the final projection layer is $N$? Then, how is this method going to handle dynamic numbers of tokens? This is one of the main problems that the authors had to clearly present.
- The presentation must be improved. There are too many unnecessary details that are redundant. For example, most of the equations and the definition (Sec 3.1) can be simplified in a single equation or a few sentences.
- This paper probably does not compare to FlashAttention. Unless there is a solution that the authors put hardware algorithms in consideration, having an explicit attention score mask cannot be efficiently integrated into the FlashAttention [A] framework. Although FlexAttention [B] can handle arbitrary mask values, the explicit materialization of attention scores as did in this paper cannot bypass memory bandwidth overhead. FlashAttention is now being used everywhere, thus the comparisons must be based on FlashAttention.
- I cannot understand how the element-wise multiplication of M to the attention score works. Since M is not constrained to be non-negative, M can completely change attention values.

[A] Dao et al. FlashAttention: Fast and Memory-Efficient Exact Attention with IO-Awareness\
[B] https://pytorch.org/blog/flexattention/

**Questions:**

- How exactly is M being generated?
- How can it handle arbitrary input lengths?
- In the explanation about Table 3, the authors mention that they augment vanilla Transformer with additional linear layers. How are these layers being added? I cannot agree with the finding that the number of parameters do not correlate with performance gains. I suspect these additional linear layers are being added in a wrong manner such as without residual connections. Instead of adding linear layers, how does the performance change if simply stacking more transformer encoder layers?
- How is this approach better than vanilla self-attention given that both dynamically generate scores? Self attention should be much precise given that it actually visits other tokens' keys, whereas the proposed method just uses linear projections per each token.

---

> ### Author Response · Authors · 2024-11-22
> **Response to the Reviewer sa9g (Part 1)**
>
> Thank you for your encouraging feedback on the algorithm's simplicity, strong benchmark results, and relevance to the multi-modality problem. We deeply appreciate your feedback and will address your concerns in detail below.
>
> ### Weaknesses
>
> 1. **The method part is not explaining the concept correctly. In Sec 3.2, which is supposed to be the core contribution of this paper, the final output of the stack of layers is in the shape of N. Do the authors mean that the output dimension of the final projection layer is ? Then, how is this method going to handle dynamic numbers of tokens? This is one of the main problems that the authors had to clearly present.**
>
> Thank you for your insightful feedback regarding Section 3.2 of our manuscript. We apologize for any confusion caused and appreciate the opportunity to clarify our method and its handling of dynamic numbers of tokens.
> In **Lines 204 to 208**, we state that the output mask produced by our Learnable Context-Aware Attention Mask (LCAAM) has dimensions of $(B, N, N)$ for self-attention and $(B, N_q, N_k)$ for cross-attention setups, where:
> - $B$ is the batch size,
> - $N$  is the number of tokens in the input sequence to the layer,
> - $N_q$ is the number of query tokens, and
> - $N_k$ is the number of key tokens.
>
> The output dimension of the final projection layer indeed corresponds to the number of tokens $N$ or $N_k$ in cross-attention scenarios). **Our method is specifically designed to handle dynamic numbers of tokens by learning attention masks that adapt to varying sequence lengths and content**. This adaptability is achieved through the learnable parameters in LCAAM, which adjust the attention distribution based on the input data.
>
> Regarding the concern about **how this method is going to handle dynamic numbers of tokens**, Transformer-based approaches typically require a fixed window size, which necessitates padding shorter sequences with zeros to align them to the context size. LCAAM addresses this by effectively interpreting such padding during its mask computation, ensuring that padded tokens do not distort attention scores. This capability was validated through experiments on the MAD dataset (audio descriptions) and QVHighlights (moment retrieval), both involving input sequences of varying lengths. The strong performance in these tasks highlights LCAAM's effectiveness in handling padded sequences and maintaining robust token prioritization across varying input sizes.
>
> 2. **The presentation must be improved. There are too many unnecessary details that are redundant. For example, most of the equations and the definition (Sec 3.1) can be simplified in a single equation or a few sentences.**
>
> Thank you for your thoughtful feedback. We will incorporate the suggested changes in the camera-ready version by simplifying the equations and definitions.
>
> 3. **This paper probably does not compare to FlashAttention. Unless there is a solution that the authors put hardware algorithms in consideration, having an explicit attention score mask cannot be efficiently integrated into the FlashAttention [A] framework. Although FlexAttention [B] can handle arbitrary mask values, the explicit materialization of attention scores as did in this paper cannot bypass memory bandwidth overhead. FlashAttention is now being used everywhere, thus the comparisons must be based on FlashAttention.**
>
> We appreciate the reviewer’s insightful feedback regarding the comparison with FlashAttention. **Our proposed Local Context-Aware Attention Mechanism (LCAAM) is designed to complement, rather than compete with, existing frameworks like FlashAttention and FlexAttention**. LCAAM can be seamlessly integrated with FlashAttention to enhance hardware efficiency by leveraging FlashAttention’s optimized memory I/O operations while introducing global context-awareness and learnable masks for prioritizing semantically significant tokens, particularly in multimodal contexts. Similarly, LCAAM can adapt to FlexAttention by augmenting its hierarchical attention and dynamic high-resolution token selection with adaptive soft-masking, thereby further improving semantic prioritization without incurring significant memory bandwidth overhead. Unlike FlashAttention, which focuses on optimizing memory and computational efficiency for long sequences, and FlexAttention, which targets reducing computational costs in high-resolution vision-language tasks, LCAAM enhances self-attention mechanisms through dynamic token prioritization across multiple modalities. This complementary functionality ensures that LCAAM not only maintains compatibility with widely adopted frameworks like FlashAttention but also extends their capabilities to support more complex multimodal tasks.

---

> ### Author Response · Authors · 2024-11-22
> **Response to the Reviewer sa9g (Part 2)**
>
> **Table 1.** Comparison betwwen LCAAM, Flash Attention and Flex Attention.
> | **Aspect**           | **LCAAM**                                                                 | **FlexAttention**                                                                 | **FlashAttention**                                                       |
> |----------------------|---------------------------------------------------------------------------|------------------------------------------------------------------------------------|--------------------------------------------------------------------------|
> | **Core Innovation**  | Learnable masks for context-aware token prioritization in multimodal tasks. | Hierarchical attention with dynamic high-resolution token selection for vision-language tasks. | IO-efficient exact attention leveraging GPU tiling to reduce memory overhead for long sequences. |
> | **Primary Focus**    | Enhances semantic and contextual alignment across modalities.             | Reduces computational cost for processing high-resolution image tokens.            | Accelerates self-attention in long-sequence tasks with reduced memory footprint.                |
> | **Token Interaction**| Learn global relationships dynamically across all tokens.                  | Combines low- and high-resolution image tokens for selective attention.              | Retains exact token relationships but optimizes memory and computation.                        |
> | **Applicability**    | Multimodal tasks (e.g., text, audio, video) requiring cross-modal attention. | Vision-heavy tasks (e.g., high-resolution VQA, image-based reasoning).              | Long-sequence NLP or tasks with large input sizes (e.g., GPT, BERT).                          |
> | **Computational Focus** | Adds computational overhead for token prioritization but supports diverse inputs. | Reduces cost by limiting high-resolution token processing.                        | Optimizes speed and memory usage for attention-heavy workloads.                                |
> | **Strengths**        | Improves multimodal alignment with semantic prioritization.               | Efficiently balances low- and high-resolution image processing.                      | Offers significant speedup and memory efficiency for large input sequences.                     |
> | **Weaknesses**       | Overhead may impact scalability in very long sequences.                   | Limited generalizability outside vision-language domains.                           | Lacks support for semantic or contextual prioritization of tokens.                            |

---

> ### Author Response · Authors · 2024-11-22
> **Response to the Reviewer sa9g (Part 3)**
>
> 4. **I cannot understand how the element-wise multiplication of M to the attention score works. Since M is not constrained to be non-negative, M can completely change attention values.**
>
> The key insight of the proposed method is the design of the mask $M$, generated by the LCAAM module, through a ReLU-activated architecture (see Section 3.2). This ensures non-negative outputs from intermediate layers, except during the final mask generation step, which maps to the attention space. While the final layer is unconstrained in sign, it aligns contextually with attention scores, enabling \( M \) to function as a soft filter. Instead of directly zeroing out or altering negative values, $M$ adjusts attention scores more subtly.
>
> Building on SwinBERT's **soft-masking strategy**, the proposed approach applies $M$ **before the softmax operation**, inherently constraining attention values to be non-negative, as softmax produces only positive probabilities. This not only preserves well-defined attention scores but also refines the attention weight distributions, as illustrated in **Figures S5 and S7 (Appendix)**. By prioritizing context-aware token selection, this strategy enhances performance, particularly in tasks like audio description, where nuanced attention is critical.
>
> ### Questions
>
> 1.  **How exactly is M being generated?**
>
> The mask $M$ is computed using a sequence of linear layers followed by a ReLU activation, as detailed in Equations (2)–(4). This approach ensures $M$ is contextually generated for any input length $N$, dynamically leveraging token embeddings to determine relationships. Unlike static mechanisms, LCAAM's learnability adapts to sequences of varying lengths due to its projection-based design.
>
> 2. **How can it handle arbitrary input lengths?**
>
> All Transformer-based approaches require a fixed window size, which necessitates padding shorter sequences with zeros to align them to the context size. LCAAM effectively interprets such padding during its mask computation, ensuring that these padded tokens do not distort attention scores. This capability was rigorously tested during experiments on the **MAD dataset (audio descriptions)** and the **QVHighlights (Moment retrieval)**, both of which involved input sequences of varying lengths. The strong performance achieved in these tasks demonstrates LCAAM's effectiveness in handling padded sequences and ensuring robust token prioritization across different input sizes.
>
> 3. **In the explanation about Table 3, the authors mention that they augment vanilla Transformer with additional linear layers. How are these layers being added? I cannot agree with the finding that the number of parameters do not correlate with performance gains. I suspect these additional linear layers are being added in a wrong manner such as without residual connections. Instead of adding linear layers, how does the performance change if simply stacking more transformer encoder layers?**
>
> Thank you for the opportunity to clarify how we constructed the **"Full Attention w/ same number params".** model. The linear layers were integrated into the Transformer by augmenting the output of each Multi-Head Attention (MHA) module with additional linear transformations, applied uniformly across all Transformer layers. This design choice ensures a parameter-equivalent comparison with the LCAAM configuration and avoids introducing unnecessary complexity that could confound results. Regarding the parameter-performance relationship, the experiments demonstrate that merely increasing the number of parameters, such as through additional linear layers, does not consistently yield performance gains and can lead to overfitting. This highlights the importance of strategically designed mechanisms like LCAAM to enhance model learning effectively, as evidenced by its superior performance.

---

> ### Author Response · Authors · 2024-11-22
> **Response to the Reviewer sa9g (Part 4)**
>
> 4. **How is this approach better than vanilla self-attention given that both dynamically generate scores? Self attention should be much more precise given that it actually visits other tokens' keys, whereas the proposed method just uses linear projections per each token.**
>
> The mask, $\mathbf{M}$, generated by the LCAAM module refines the attention scores of vanilla self-attention, as detailed in Section 3 and illustrated in Figure 2. While Vanilla self-attention effectively captures precise token relationships, it lacks the ability to adjust these relationships based on higher-level contextual relevance. LCAAM overcomes this limitation by introducing a dynamic, learnable mechanism that prioritizes token interactions according to their contextual importance. This capability is particularly impactful in multimodal tasks, such as video-audio alignment in the MAD dataset, where intermodal associations benefit from adaptive prioritization. **Experimental results confirm that LCAAM outperforms baseline self-attention models in these scenarios.**
>
> **Figures S5 and S7 in the Appendix** illustrate the impact of incorporating the LCAAM module. **LCAAM significantly influences the distribution of attention weights by dynamically reducing their magnitude across deeper layers, with many weights approaching zero**. This behavior introduces sparsity, enhances computational efficiency, and enables the model to focus on the most relevant token relationships. In contrast, vanilla self-attention distributes weights more uniformly, lacking LCAAM's selective refinement. These findings underscore LCAAM’s advantages in tasks requiring both contextual relevance and efficiency, such as those in the MAD dataset.

---

> > ### Comment · Reviewer_sa9g · 2024-11-24
> > **Response to the Authors**
> >
> > Thank you for the detailed response. Here are my additional comments.
> >
> > 1. I believe it is over-claiming to say controlling the context length by adding attention mask is able to handle "dynamic" numbers of tokens. What the authors are proposing is highly related to MLP-Mixer [C], which has a fixed context length. Both MLP-Mixer and what the authors are proposing cannot be extended to arbitrary sequence lengths if the number of input tokens is greater than what the model is trained on. Therefore, this paper is a fixed-size method.
> >
> > 2. Although the authors provide intuitions of why this could be helpful for cases like multi-modal, I believe the authors should have evaluated their method on more general domains such as auto-regressive model training given that LCAAM is a method that's applicable to any transformer-based architectures.
> >
> > 3. Since the number of parameters is extremely important for a fair comparison, I wonder if it is being controlled. Is the paper using more memory due to the LCAAM layer? What's the parameter count for each method?
> >
> > 4. The table the authors provided is misleading, especially the descriptions about FlexAttention are wrong. Also, my point wasn't that LCAAM should compete with either FlashAttention or FlexAttention. Rather, I was wondering if it could be integrated to such methods. The authors say that it could be seamlessly integrated to FlashAttention in with very high-level explanations. However, unless there is a detailed explanation, I believe it is an over-clam.
> >
> > 5. I am not convinced with the authors' response about the multiplication between LCAAM scores and attention scores. I do completely understand that the softmax function assures non-negativity. However, what I am particularly pointing out is the multiplication to QK^T. Say a certain QK^T index has a very high value indicating high importance. By multiplying a negative value to it totally corrupts its original value, making it near zero when taken after softmax.
> >
> > 6. Instead of adding linear layers, how does the performance change if stacking more transformer encoder layers?
> >
> > [C] MLP-Mixer: An all-MLP Architecture for Vision

---

> ### Author Response · Authors · 2024-11-27
> **New Response to the reviewer sa9g (Part 1)**
>
> Q1. **I believe it is over-claiming to say controlling the context length by adding attention mask is able to handle "dynamic" numbers of tokens. What the authors are proposing is highly related to MLP-Mixer [C], which has a fixed context length. Both MLP-Mixer and what the authors are proposing cannot be extended to arbitrary sequence lengths if the number of input tokens is greater than what the model is trained on. Therefore, this paper is a fixed-size method.**
>
> We want to first clarify that we have not claimed our method can handle dynamic numbers of tokens. Our method can be integrated into any Transformer architecture, and Transformer-based models typically process a fixed number of tokens using a padding strategy.
> Moreover, MLP-Mixer and LCAAM differ fundamentally, particularly in their use of Transformers. MLP-Mixer is explicitly designed without convolutional or self-attention mechanisms and does not use Transformer architectures. Instead, it processes data entirely through MLPs that separately mix spatial and channel information in image patches, presenting a simpler alternative to attention-based models. In contrast, LCAAM operates exclusively within Transformer architectures, leveraging MLP layers with ReLU activations and softmax to refine self-attention maps by prioritizing tokens within a fixed window size. This distinction clarifies that MLP-Mixer avoids the Transformer framework entirely, while LCAAM enhances and integrates into it, targeting tasks requiring refined attention mechanisms in multimodal sequential data.
>
> Q2.**Although the authors provide intuitions of why this could be helpful for cases like multi-modal, I believe the authors should have evaluated their method on more general domains such as auto-regressive model training given that LCAAM is a method that's applicable to any transformer-based architectures.**
>
> We kindly invite the reviewer to refer to Table 1(a), which provides a clear example of the **autoregressive task (Audio Description Generation).** **This table highlights the performance of LLaMA 7B integrated with LLaMA Adapter layers, both with and without the inclusion of LCAAM**. Table 1(a) highlights a notable maximum improvement of 12.7 points for the R@5/16 metric, alongside an average improvement of 8.23 points across all metrics. This demonstrates the effectiveness of the approach, yielding significant benefits even in autoregressive tasks.
>
> Q3.**Since the number of parameters is extremely important for a fair comparison, I wonder if it is being controlled. Is the paper using more memory due to the LCAAM layer? What's the parameter count for each method?**
>
> **In Table 4, the computational overhead of each model is measured using FLOPs (Floating-Point Operations), MACs (Multiply-Accumulate Operations), and parameter count**. FLOPs indicate the total number of floating-point calculations, while MACs represent the multiplications and additions performed by the model. Parameter count reflects the number of learnable weights in the model. These metrics provide a basis for comparing the computational cost of different models.

---

> ### Author Response · Authors · 2024-11-27
> **New Response to the reviewer sa9g (Part 2)**
>
> Q4. **The table the authors provided is misleading, especially the descriptions about FlexAttention are wrong. Also, my point wasn't that LCAAM should compete with either FlashAttention or FlexAttention. Rather, I was wondering if it could be integrated to such methods. The authors say that it could be seamlessly integrated to FlashAttention in with very high-level explanations. However, unless there is a detailed explanation, I believe it is an over-clam.**
>
> The Flex Attention method, as described in the paper, is designed to integrate into most vision-language models. It processes high-resolution images by downsampling them to create low-resolution tokens, while also encoding high-resolution tokens. Initially, only low-resolution image and text tokens are used for efficiency, with high-resolution tokens introduced later to capture finer details. The method includes a feature selection module to choose significant tokens and a hierarchical self-attention module to integrate them into the hidden states, culminating in a linear projector that generates the textual output. These details are consistent with what we discussed in the Table in our response
>
> **We provide details below on how to integrate our LCAAM with FlashAttention:**
>
> For users working with PyTorch version 2.2 or later, FlashAttention is natively integrated into the `scaled_dot_product_attention` function. Here's the relevant function implementation for clarity:
>
> ```
> # Efficient implementation with LCAAM
> def scaled_dot_product_attention(query, key, value, attn_mask=None, dropout_p=0.0, is_causal=False, scale=None, LCAAM_soft_mask=None) -> torch.Tensor:
>     L, S = query.size(-2), key.size(-2)
>     scale_factor = 1 / math.sqrt(query.size(-1)) if scale is None else scale
>     attn_bias = torch.zeros(L, S, dtype=query.dtype)
>
>     if is_causal:
>         assert attn_mask is None
>         temp_mask = torch.ones(L, S, dtype=torch.bool).tril(diagonal=0)
>         attn_bias.masked_fill_(~temp_mask, float("-inf"))
>         attn_bias.to(query.dtype)
>
>
>     if attn_mask is not None:
>         if attn_mask.dtype == torch.bool:
>             attn_bias.masked_fill_(~attn_mask, float("-inf"))
>         else:
>             attn_bias += attn_mask
>
>
>     attn_weight = query @ key.transpose(-2, -1) * scale_factor
>     attn_weight += attn_bias
>
>
>     # Apply the LCAAM soft mask if provided
>     if LCAAM_soft_mask is not None:
>         assert LCAAM_soft_mask.shape == attn_weight.shape, "LCAAM_soft_mask must match the shape of attn_weight"
>         attn_weight = attn_weight * LCAAM_soft_mask
>
>
>     attn_weight = torch.softmax(attn_weight, dim=-1)
>     attn_weight = torch.dropout(attn_weight, dropout_p, train=True)
>     return attn_weight @ value
> ```
>
> To incorporate LCAAM to FlashAttention, the adjustment can be inserted just before the `torch.softmax` operation. Multiply the `attn_weight` with the output of LCAAM (  LCAAM_soft_mask) for enhanced token prioritization. Additionally, enabling FlashAttention is as simple as activating `torch.backends.cuda.enable_flash_sdp()`.
>
> **Please find more details at**: https://pytorch.org/docs/2.2/generated/torch.nn.functional.scaled_dot_product_attention.html

---

> ### Author Response · Authors · 2024-11-27
> **New Response to the reviewer sa9g (Part 3)**
>
> Q5. **I am not convinced with the authors' response about the multiplication between LCAAM scores and attention scores. I do completely understand that the softmax function assures non-negativity. However, what I am particularly pointing out is the multiplication to QK^T. Say a certain QK^T index has a very high value indicating high importance. By multiplying a negative value to it totally corrupts its original value, making it near zero when taken after softmax.**
>
> Our approach was grounded in the empirical observation, as noted in **lines 049–053**, that **not all tokens in complex input sequences contribute equally to the final output**. Building on this insight, we developed LCAAM, a module designed to analyze entire sequences by considering the complete context window at once. This method provides a **broader, more global perspective compared to the original attention scoring mechanism**, potentially enabling a more comprehensive understanding.
>
> **Figure S5** illustrates this phenomenon, showing a noticeable skew in attention scores toward the left (many scores being zero), particularly when analyzing batches of 64 samples. Furthermore, this skew becomes more pronounced in later layers of the model. For instance, comparing the 1st layer to the 8th layer reveals that attention scores in the 8th layer are even more biased to the left, with a higher proportion of zeros. This indicates that as the model progresses through layers, **it increasingly prioritizes a smaller subset of tokens by making the mask values negative, thereby driving the final attention values closer to zero and focusing its attention more selectively on those most critical to prediction**.
>
> This trend underscores the model’s ability to refine its focus as it processes deeper representations of the input sequence. It aligns with the results in **Table 1**, demonstrating that using fewer tokens does not diminish performance but rather emphasizes the importance of relevant tokens. However, as with any deep learning model, limitations remain, and we invite the reviewer to revisit **Figure S8** for an example illustrating these challenges.

---

> ### Author Response · Authors · 2024-11-27
> **New Response to the reviewer sa9g (Part 4)**
>
> Q6. **Instead of adding linear layers, how does the performance change if stacking more transformer encoder layers?**
>
> Thank you for suggesting the experiment. We conducted it by modifying the baseline model described in **Table 3**. Specifically, we enhanced the audiovisual encoder used for AD generation by stacking more Transformer encoder layers, bringing the total parameter count closer to that of Multi-LCAAM. This new baseline model reached approximately **7.180 billion parameters**, compared to **6.930 billion** for the baseline and **7.072 billion** for Multi-LCAAM. After training the new baseline model, we evaluated its performance on the validation subset using the CIDEr and Rouge-L metrics. The modified model achieved a Rouge-L score of $13.806$ and a CIDEr score of $16.223$, demonstrating improved performance over the baseline. However, these results still fell short of the effectiveness achieved by Multi-LCAAM.
>
> **Table. Performance of Baseline Models on the Validation Subset for the AD Generation Task.**
> The new baseline (*) stacks transformer layers to approximate the parameter count of the Multi-LCAAM model, achieving competitive results.
>
> | Model           | R-L (↑) | C (↑)   | Params (↓)  |
> |------------------|---------|---------|-------------|
> | Baseline        | 12.92   | 15.46   | 6.930 B     |
> | New Baseline (*)| 13.806  | 16.223  | ~7.180 B    |
> | Multi-LCAAM     | 14.28   | 17.11   | 7.072 B     |

---

> ### Comment · Reviewer_sa9g · 2024-11-27
> **Response to the Authors**
>
> Thank you for the authors' response. Here, I want to further clarify my concerns.
>
> ----
>
> I do not want the main focus to be on implementation details, but I believe I should correct these false claims.
>
> I am very confused why the authors are keep on claiming that FlexAttention is a vision-related model. FlexAttention is just a new GPU kernel provided by PyTorch, not a paper. I believe the authors are wrongly referring to "FlexAttention for Efficient High-Resolution Vision-Language Models", which comes out on top when one searchs for "Flex Attention" on Google. However, I am referring to the link I provided in my very initial review.
>
> The authors' response on the implementation side that LCAAM is being integrated into FlashAttention is wrong. I wonder if (A) the code that the authors provided is being directly used for their implementation, or if (B) it is provided as pseudo code.
> If (A), it is literally using naive PyTorch, not any of GPU kernels.
> If (B), the claim is still wrong. If using attn_mask , PyTorch does not use FlashAttention. Instead, it uses memory efficient attention from xFormers.
> https://github.com/pytorch/pytorch/blob/41e218984359a0dc93ee54776bba792b69b17b19/aten/src/ATen/native/transformers/attention.cpp#L730-L740
>
> Even if one forces to integrate LCAAM into Flash Attention, the materialization of attention scores does not accord with the algorithm of the kernel.
>
> Given the authors are wrongly understanding the implementation details, I believe the time-related metrics (e.g., one in Table 4) are not measured using FlashAttention, but pure PyTorch.
>
> I think there isn't a simple way to integrate LCAAM into either FlashAttention or FlexAttention. The authors could have just admitted it, but these false claims are making me extremely hard to trust the other answers as well.
>
> Please revisit the PyTorch documentation link that the authors provided and the source code above.
>
> ----
>
> Statistical significance of the experiments. Are these results average of multiple runs, or max values (if so, how many specifically)? What's std of the results?
>
> ----
>
> The authors' previous response was **"Our method is specifically designed to handle dynamic numbers of tokens by learning attention masks that adapt to varying sequence lengths and content."**, and I want it to be clear that it **cannot** handle dynamic numbers of tokens.
> The reason I am mentioning MLP-Mixer is not because of it's overall architectural design, but specifically because of the mixer matrix, and was trying to make it clear that LCAAM is a fixed-size method.
> Self Attention is $\textbf{MV}$ where $\textbf{QK}^\intercal = \textbf{M} \in \mathbb{R}^{N \times N}$. MLP-Mixer can also be viewed as $\textbf{MV}$ where $\textbf{M}$ is a fixed-size trainable matrix. What I meant is LCAAM is just as MLP-Mixer with fixed-size mixer matrix, and the difference is it is just being generated from input data.
>
> ----
>
> The response about the multiplication still does not mathematically explain why it works.
> Say there are tokens that should not have high attention scores. Should $\textbf{QK}^\intercal$ values of the corresponding tokens be very small negative values? Then what if corresponding $\textbf{M}$ values are also negative?
> Otherwise, should the $\textbf{QK}^\intercal$ values be positive? Again, then what if corresponding $\textbf{M}$ values are also positive?
> Unless there is a mathematical guarantee, hoping for the model to magically optimize does not make sense.
>
> So the mechanism of LCAAM is making each token predict attention bias not only for itself but also for all other $N-1$ tokens as well. This could work for settings where $N$ is fairly small, but it is certainly predictable that this ambiguity will lead to extreme instabilities if $N$ becomes larger, such as for general LLM cases with 1K~8K (or even more) sequence length.
>
> A question on top of the previous comment; what's $N$ for all datasets that this paper is evaluating on? I can see one dataset uses the sequence length of 50 tokens only, and I believe $N$ is extremely small for all of these benchmarks (mostly smaller than 1K?).
>
> ----
>
> Is LCAAM being added to all transformer layers, or added to a subset of layers (if so, specifically where)?

---

> > ### Author Response · Authors · 2024-12-01
> > **Response to Reviewer sa9g (PART 1)**
> >
> > ---
> >
> > **FlexAttention and Flash Attention**
> >
> > **Flex Attention:**
> > Based on the PyTorch documentation on FlexAttention (https://pytorch.org/blog/flexattention/), the `score_mod` function is used to compute a modified dot product of a query token and a key token. The additional arguments specify the context of the computation: `b` represents the current element in the batch, `h` refers to the current attention head, `q_idx` indicates the position in the query, and `kv_idx` corresponds to the position in the key/value tensors. This logic is exemplified in the following loop:
> >
> > ```
> > for b in range(batch_size):
> >     for h in range(num_heads):
> >         for q_idx in range(sequence_length):
> >             for kv_idx in range(sequence_length):
> >                 modified_scores[b, h, q_idx, kv_idx] = score_mod(scores[b, h, q_idx, kv_idx], b, h, q_idx, kv_idx)
> > ```
> >
> > While this implementation focuses on optimizing attention calculations, it is essential to recognize its limited scope. This approach is designed primarily for computational efficiency and does not align with the objectives or methodologies of LCAAM. Our focus lies in **Token Prioritization**, which is entirely absent in FlexAttention's implementation.
> > For a more meaningful comparison or application, we recommend adapting LCAAM to modify the QK^T product within the softmax operation. This would align with the concept of `score_mod` in FlexAttention but integrates token prioritization by applying our soft masking strategy. **Each element of the masking can be multiplied with the scores (score_mod) in a similar nested loop structure, ensuring our module's benefits are incorporated effectively**.
> >
> >
> > **Flash Attention:**
> > We’ve added pseudocode to address your question: “Rather, I was wondering if it could be integrated to such methods. The authors say that it could be seamlessly integrated to FlashAttention with very high-level explanations.”
> > To avoid any confusion, we want to clarify that we did not use FlashAttention in our experiments. As we have explained, **the scopes of FlashAttention and LCAAM are completely different**:
> > - FlashAttention is a **code optimization**.
> > - LCAAM focuses on **input sequence analysis and token prioritization**.
> >
> >
> > **We also want to highlight a few points**:
> >
> > - The link the reviewer provided includes the implementation of FlashAttention, which can be accessed here: https://github.com/pytorch/pytorch/blob/41e218984359a0dc93ee54776bba792b69b17b19/aten/src/ATen/native/transformers/attention.cpp#L713-L729.
> > - The SDPA backend leverages Flash Attention and is officially supported, as outlined in the PyTorch documentation: https://pytorch.org/docs/stable/generated/torch.nn.attention.SDPBackend.html#torch.nn.attention.SDPBackend. The same GitHub code you referred to corresponds to EFFICIENT_ATTENTION, which is the efficient attention backend for Scaled Dot-Product Attention. However, in the same documentation, the FLASH_ATTENTION option can be found, which is distinct from the implementation we referenced earlier.
> > - Our approach employs a soft masking mechanism derived from LCAAM. Unlike the conventional binary mask of 0s and 1s (attn_mask), our original implementation required modifications to the Transformer class to accommodate this approach.
> > - **At no point in our paper do we claim to use Flash Attention**. It is important to note that LCAAM introduces computational overhead, making it reasonable to avoid using any code optimizations when evaluating its impact. This ensures an accurate assessment of LCAAM's influence on the overall implementation. **Any subsequent code modifications are purely optimizations, representing implementation and engineering improvements rather than contributions to the core research**. As we initially responded, our proposed LCAAM is designed to complement, rather than compete with, existing optimization frameworks like FlashAttention and FlexAttention.

---

> > > ### Author Response · Authors · 2024-12-01
> > > **Response to Reviewer sa9g (PART 2)**
> > >
> > > ---
> > > **MLP-MIXER and context window**
> > >
> > > To clarify the sentence: **"Our method is specifically designed to handle dynamic numbers of tokens by learning attention masks that adapt to varying sequence lengths and content”**, it highlights that LCAAM is capable of selecting multiple tokens as "important". This adaptability is evident in Figure S5, which shows that the **number of tokens deemed "important" (non-zeros) changes across different Transformer layers** (i.e. the number of important tokens changes dynamically per each transformer layer). Additionally, the tasks listed in **Table 1 involve varying sequence sizes, with each task having its own specific sequence and context size**. However, **this does not imply that the model can process different sequence sizes within the same forward pass**. This also was already addressed in our previous response: **"Our method can be integrated into any Transformer architecture, and Transformer-based models typically process a fixed number of tokens using a padding strategy”**.
> > >
> > >
> > > Thanks for mentioning MLP-Mixer. While we appreciate the comparison, we respectfully disagree with the statement that **LCAAM is just as MLP-Mixer with fixed-size mixer matrix, and the difference is it is just being generated from input data.** There are clear distinctions between our proposed LCAAM and MLP-Mixer.
> > >
> > > MLP-Mixer contains two distinct types of layers: one employs MLPs applied independently to individual image patches, and the other applies MLPs across patches, facilitating the "mixing" of spatial information. In contrast, **LCAAM processes the entire sequence globally using MLPs and ReLU layers (absent in MLP-Mixer), enabling a holistic consideration of the global sequence rather than focusing on localized patch-level interactions**. Moreover, LCAAM incorporates attention scores and soft masking operations to dynamically prioritize tokens based on their contextual significance, a feature not present in MLP-Mixer.
> > >
> > > ---
> > > **The multiplication still does not mathematically explain why it works.**
> > >
> > > The element-wise multiplication between $\mathbf{QK}^\intercal$ and $\mathbf{M}$ in LCAAM works because it allows the raw token similarities $\mathbf{QK}^\intercal$, which encode pairwise relationships, to be dynamically modulated by the learned contextual mask $\mathbf{M}$. For instance, if $\mathbf{QK}^\intercal[i,j]$ represents the similarity between token $i$ and token $j$, then $\mathbf{M}[i,j]$ scales this value based on the global contextual importance of the token pair.
> > > When $\mathbf{QK}^\intercal[i,j] > 0$, $\mathbf{M}[i,j]$ is optimized to either amplify this value if the relationship is significant or reduce it if not. Conversely, if $\mathbf{QK}^\intercal[i,j] < 0$ (indicating dissimilarity) and the relationship is not significant, $\mathbf{M}[i,j]$ further reduces the impact further by driving the softmax of the product $\mathbf{QK}^\intercal[i,j] \cdot \mathbf{M}[i,j]$ closer to zero. This ensures that dissimilar tokens do not gain undue attention.
> > > The multiplication works mathematically because $\mathbf{M}$ is learned such that it aligns with the **task objective, reinforcing desirable attention patterns and suppressing irrelevant ones**. **The training process simultaneously optimizes $\mathbf{QK}^\intercal$ and $\mathbf{M}$ to ensure their product respects task-specific relationships**, preserving pairwise structure while dynamically scaling attention based on learned priorities.
> > >
> > > ---
> > > **What's $N$ for all datasets that this paper is evaluating on?**
> > >
> > > **Table. Sequence Lengths and Attention Modes Across Tasks and Datasets.** The sequence lengths for the tasks outlined in the **Table 1 of the main paper**, are presented here, along with a detailed comparison of attention modes (cross or self-attention) and their respective sequence lengths.
> > >
> > >
> > > | Task                  | Dataset         | Sequence Length      | Attention Mode     |
> > > |-----------------------|-----------------|----------------------|--------------------|
> > > | AD                   | Mav2            | $N = 50$           | Self Attention     |
> > > | Moment Retrieval     | QVHighlights    | $N_q = 75, N_k = 32$ | Cross Attention    |
> > > | Highlights Detection | QVHighlights    | $N_q = 75, N_k = 32$ | Cross Attention    |
> > > | Image Classification | ImageNet1k      | $N = 197$         | Self Attention     |
> > > | Video Captioning     | MSRVTT          | $N = 834$          | Self Attention     |

---

> > > > ### Author Response · Authors · 2024-12-01
> > > > **Response to Reviewer sa9g (PART 3)**
> > > >
> > > > ---
> > > > **Is LCAAM being added to all transformer layers, or added to a subset of layers (if so, specifically where)?**
> > > >
> > > > As outlined in **lines 213 to 215**, the mask can be applied in two distinct ways: **globally across all transformer layers in the stack (LCAAM, Section 3.2) or individually for each layer (Multi-LCAAM, Section 3.4)**.
> > > > When applied globally, we added LCAAM to each Transformer layer, with all parameters shared across the layers. In this setup, a single mask is consistently used across all transformer layers, adhering to the standard masking approach in Attention layers.
> > > > In contrast,  the Multi-LCAAM setup, detailed in Section 3.4, consists of a series of $L$ Transformer layers. **Each layer $l$ is equipped with its own unique, learnable mask, denoted as $\mathbf{M}_l$**. These masks, $\mathbf{M}_l$, are **independently learned for each layer**, allowing each one to adapt its masking operation to the specific representation it processes. With the output of one layer feeding into the next, this setup facilitates a hierarchical representation learning process across the stack, enabling increasingly refined representations at each level.

---

> > > > > ### Author Response · Authors · 2024-12-03
> > > > > **Response to Reviewer sa9g (PART 4)**
> > > > >
> > > > > ---
> > > > > **Statistical significance of the experiments. Are these results average of multiple runs, or max values (if so, how many specifically)? What's std of the results?**
> > > > >
> > > > > **Table 1.: Comparison of performance metrics using different random seeds.** The table presents a detailed comparison of evaluation metrics (R-L: ROUGE-L, C: CIDEr, R@5/16: Recall at top-5/16) between the LLaMA Adapter and our proposed method **(extended version of Table 1(a) in the paper)**. **Results are reported as mean ± standard deviation**, based on experiments conducted with multiple random seeds and a fixed temperature setting of 0.
> > > > >
> > > > >
> > > > > | Method          | R-L  | C   | R@5/16  |
> > > > > |------------------|-------------------|-----------------|---------------------|
> > > > > | LlaMA Adapter   | 10.0 (± 0.65)     | 9.0 (± 0.35)    | 42.86 (± 0.55)      |
> > > > > | **Ours**            | **13.54 (± 0.5)**     | **18.56 (± 0.2)**   | **56.15 (± 0.3)**       |
> > > > >
> > > > >
> > > > > The results clearly illustrate that our method consistently outperforms the LlaMA Adapter across all metrics evaluated. For Rouge-L (R-L), the mean score achieved by our method is 13.54, compared to the LlaMA Adapter's 10.0, with respective standard deviations of 0.5 and 0.65. This indicates not only a higher average performance but also greater consistency. In the CIDEr (C) metric, our method more than doubles the average score, achieving 18.56 compared to the LlaMA Adapter's 9.0, while also **demonstrating reduced variability** (0.2 vs. 0.35). Similarly, for recall at top-5/16 (R@5/16), our method attains a significantly higher average of 56.15 compared to 42.86, with lower standard deviation values of 0.3 versus 0.55.
> > > > >
> > > > > **For Tables 1(b) and 1(c) in the main paper**, the reported values represent the averages obtained from multiple experiments using different seed values. To provide additional clarity, we now include both the averages and their corresponding standard deviations:
> > > > >
> > > > > **Table 2. Moment Retrieval and Highlight Detection Performance in QVHighlights.**Performance of models on Moment Retrieval (MR) and Highlight Detection (HD) tasks across multiple seeds, evaluated using R1, mAP, IoU@0.7, Avg mAP, and HIT@1 metrics. **Results are reported as mean ± standard deviation.**
> > > > >
> > > > >
> > > > > | Model     | MR R1 IoU@0.7 | MR mAP Avg  | HD >= Very Good mAP  | HD >= Very Good HIT@1 |
> > > > > |-----------|-----------------------|-------------------|----------------------------|------------------------------|
> > > > > | QD-DETR   | 44.98 (± 0.8)          | 39.86 (± 0.6)      | 38.94 (± 0.4)               | 62.40 (± 1.4)                 |
> > > > > | **Ours**      | **46.94 (± 0.6)**          | **42.32 (± 0.6)**      | **39.70 (± 1.0)**              | **63.33 (± 0.8)**                 |
> > > > >
> > > > >
> > > > > **We sincerely appreciate your valuable feedback and will incorporate this information for the remaining experiments in Table 1 into the camera-ready version to enhance clarity.**

---

> > > > > > ### Author Response · Authors · 2024-12-04
> > > > > > **Response to Reviewer sa9g (PART 5)**
> > > > > >
> > > > > > ---
> > > > > > **Flash Attention V2 and LCAAM**
> > > > > >
> > > > > > **Table: Performance Comparison Between Multi-LCAAM and Multi-LCAAM with Flash Attention V2**. The table highlights that LCAAM can be effectively implemented using FlashAttention V2. While our research primarily focused on performance gains, the use of FlashAttentionV2 also demonstrates potential for reducing computational resource usage.
> > > > > >
> > > > > > | Model                         | RL       | C          | FLOPs       | MACs       | Params   | Latency  |
> > > > > > |-------------------------------|----------|------------|-------------|------------|----------|----------|
> > > > > > | Multi-LCAAM                  | 14.28    | 17.11      | 3.43 TFLOPs | 1.71 TMACs | 7.072 B  | 88.60 ms |
> > > > > > | Multi-LCAAM + Flash Attention V2 | 14.1828 | 17.23122  | **2.272 TFLOPs** | **1.13 TMACs**| 7.072 B  | **64.30 ms** |
> > > > > >
> > > > > > The comparison between Multi-LCAAM and Multi-LCAAM with Flash Attention V2 demonstrates that while the RL (Rouge-L) and C (CIDEr) scores remain within marginally similar ranges, indicating minimal change in these performance metrics, significant improvements are observed in computational efficiency. The FLOPs and MACs are reduced substantially, from 3.43 TFLOPs to 2.272 TFLOPs and from 1.71 TMACs to 1.13 TMACs, respectively. Additionally, the latency sees a notable decrease from 88.60 ms to 64.30 ms, highlighting the efficiency gains achieved with Flash Attention V2. This indicates that the integration of Flash Attention V2 enhances the model's computational performance without compromising output quality.

---

### Official Review · Reviewer_MBg4 · 2024-11-03

**Soundness:** 3
**Presentation:** 3
**Contribution:** 3
**Rating:** 6
**Confidence:** 4

**Summary:**

This work proposes a learnable attention mask to soften the attention map for cross-modal understanding. The authors claim that due to semantic density, the original attention mechanism may not fully capture the relation between different modalities.  Therefore, they propose the learnable context-aware attention mask (LCAAM) to regulate and prioritize the token importance of the long sequence. The implementation of LCAAM is a stack of MLP layers. The authors provide extensive experiments on various task settings and benchmarks.

**Strengths:**

1. Due to the distinction of semantic density, there is a mismatch of token granularity between different input modalities. Hence, the motivation of this work, regulating the token importance of different modalities, makes sense to me.
2. I believe the implementation of LCAAM (line 224-226) is easy to follow.
3. The extensive experiments demonstrate the effectiveness of this work. The ablation study is comprehensive.

**Weaknesses:**

1. Although the visualization in the A.4.3 illustrates the learned context, it is not intuitive that the proposed module can learn the context along the sequence.
2. I think LCAAM is related to STN [1], which produces a matrix to regularize the convolution feature.


[1] Spatial Transformer Networks. NIPS 2015

**Questions:**

1. What is the benefit of LCAAM on language dataset? It seems like it cannot be generalized to NLP since the semantic density of language is quite high and the context mask may not help.
2. I believe this technique would be helpful for text2vedio generation. I hope the author can discuss it.

---

> ### Author Response · Authors · 2024-11-22
> **Response to the Reviewer MBg4**
>
> Thank you for the insightful feedback and for highlighting both the strengths and the areas that require further clarification in our work. We appreciate your detailed comments and will address the noted weaknesses and questions below.
>
> ### Weaknesses
>
> 1. **Although the visualization in the A.4.3 illustrates the learned context, it is not intuitive that the proposed module can learn the context along the sequence:**
>
> The Qualitative figures illustrate the outputs of the LCAAM module's first Transformer layer, highlighting how it learns to focus dynamically on relevant visual and audio tokens along the sequence. The x-axis represents video tokens (indices 0–24) and audio tokens (indices 25–49), enabling a direct comparison of their attention patterns. Heatmaps show how the model assigns higher attention to tokens that align with the narrative context, such as focusing on static visual elements (e.g., outdoor scenes in **Figure S8**) while minimizing unrelated audio tokens, or emphasizing dynamic scenes like actions in a maize field alongside corresponding sounds (e.g., crunching or silence in Figure **S9**). This evolving attention distribution reflects LCAAM's ability to adapt and selectively attend to meaningful transitions in the audio-visual sequence.
>
> 2. **I think LCAAM is related to STN [1], which produces a matrix to regularize the convolution feature.:**
>
> LCAAM differs fundamentally from STN in scope and application. STN focuses on spatial feature transformations to achieve invariance to geometric distortions, whereas LCAAM adjusts attention weights to dynamically prioritize token relevance in sequential data. Additionally, STN employs explicit geometric operations, while LCAAM leverages attention weight modifications to refine contextual relationships. These differences highlight LCAAM as a complementary, task-specific enhancement rather than an extension of STN principles.
>
>
> ### Questions
>
>
> 1. **What is the benefit of LCAAM on language dataset? It seems like it cannot be generalized to NLP since the semantic density of language is quite high and the context mask may not help**
>
> We appreciate the reviewer’s inquiry about LCAAM's applicability to language datasets. Our work is primarily scoped to address multimodal tasks, where token relationships span across diverse modalities, introducing complexities such as temporal alignment and cross-modal interactions. LCAAM is designed to dynamically adjust attention based on contextual significance, demonstrating significant improvements in multimodal settings like MADv2 and QVHighlights. While its principles could theoretically extend to high-semantic-density NLP tasks, such as summarization or translation, exploring these language-only domains lies beyond the scope of this study. Our focus remains on leveraging LCAAM to tackle the unique challenges posed by multimodal data, and we suggest its application to NLP as a potential area for future research.
>
> 2. **I believe this technique would be helpful for text2vedio generation. I hope the author can discuss it:**
>
> We sincerely thank the reviewer for their thoughtful suggestion regarding text-to-video generation. While our current work focuses on understanding tasks, we agree that LCAAM's dynamic attention mechanism, with its ability to prioritize cross-modal interactions, holds strong potential for bridging textual and visual representations in generation tasks. This is an exciting direction that aligns well with LCAAM's core strengths, and we look forward to exploring its application to text-to-video generation in future research.

---

> > ### Comment · Reviewer_MBg4 · 2024-11-24
> >
> > Dear authors,
> >
> > Thanks for your response.
> >
> > The motivation and contribution of this work are more clear after rebuttal. I also read other reviews. I think other reviewers may slightly misunderstand the functionality of LCAAM. Maybe it is because LCAAM is embarrassingly simple. I believe LCAAM is reproducible given its simple MLP structure. Hence, I don't find evident weakness for rejection.
> >
> > At this stage, I think this work warrants acceptance.
> >
> > Regards,

---

> ### Author Response · Authors · 2024-11-27
>
> Dear reviewer,
>
> Thank you for your kind and supportive feedback. We appreciate your understanding of our work and your recommendation for acceptance.
>
> Best regards,
>
> Authors of paper 4753

---

### Official Review · Reviewer_b6MN · 2024-11-04

**Soundness:** 3
**Presentation:** 2
**Contribution:** 2
**Rating:** 6
**Confidence:** 3

**Summary:**

This paper introduces the Learnable Context-Aware Attention Mask (LCAAM), a new mechanism designed to enhance Transformer models when dealing with complex multimodal tasks involving data like video, audio, and text. Traditional Self-Attention struggles with varying token lengths and high computational costs in such settings. LCAAM tackles this by dynamically generating a global mask that adjusts attention maps to prioritize important tokens based on their context. The authors integrate LCAAM into a BERT-like Transformer and extend it to a multi-layer setup to capture diverse information across layers. They test their approach on datasets like MADv2, QVHighlights, ImageNet-1K, and MSRVTT, showing that LCAAM improves performance and reduces redundant computations in multimodal tasks.

**Strengths:**

1. LCAAM offers a fresh take on improving attention mechanisms by dynamically adjusting attention maps based on context, particularly for multimodal data, which hasn't been extensively explored before.
2. The experiments are thorough, covering a range of datasets and tasks, from audio description generation to image classification. The ablation studies are helpful in understanding the impact of different components.
3. Addressing the challenges in multimodal data processing is important, and LCAAM shows promising improvements, suggesting it could be beneficial for advancing models in this area.

**Weaknesses:**

1. The mathematical formulation of LCAAM is a bit light. A more rigorous explanation of how the mask is generated and interacts with the attention mechanism would strengthen the paper.
2. LCAAM shows limited gains in single-modality tasks. It would be useful to investigate why this is the case and whether adjustments could improve its effectiveness in these scenarios.

**Questions:**

1. Can you delve deeper into why LCAAM improves performance? Specifically, how does the mask influence the attention distribution, and what's the theoretical basis for its effectiveness?
2. How does LCAAM stack up against other dynamic attention mechanisms like sparse attention or adaptive attention spans? Including direct comparisons would clarify its advantages.
3. Could you provide more insights into the computational overhead, such as actual latency measurements or memory usage in practical settings?

---

> ### Author Response · Authors · 2024-11-22
> **Response to the Reviewer b6MN (Part 1)**
>
> First, we want to thank you again for your thoughtful feedback, which has allowed us to clarify and refine our contributions. Below, we address the highlighted weaknesses and questions more explicitly.
>
> ### Weaknesses
>
> 1. **The mathematical formulation of LCAAM is a bit light. A more rigorous explanation of how the mask is generated and interacts with the attention mechanism would strengthen the paper**:
>
> **The Learnable Context-Aware Attention Mask (LCAAM) generates a mask $\mathbf{M}$ for token regulation in Transformers**, modeled as $\mathcal{F}(\mathbf{X}) \rightarrow \mathbf{M}$, where $\mathbf{X} \in \mathbb{R}^{B \times N \times D}$ represents the input sequence. $\mathcal{F}$ employs a stack of linear layers, with intermediate outputs:
>
> $$\mathbf{H}_1 = \text{ReLU}(\mathbf{X}\mathbf{W}_1 + \mathbf{b}_1), \quad $$
>
> $$\mathbf{H}_i = \text{ReLU}(\mathbf{H}\_{i-1} \mathbf{W}_i + \mathbf{b}_i) \text{ for } i \in \{2, \dots, L-1\},$$
>
> $$\mathbf{M} = \mathbf{H}_{L-1}\mathbf{W}_L + \mathbf{b}_L$$
>
> **The mask $\mathbf{M}$ adjusts the attention mechanism**:
> $$
> \text{Attention} = \text{softmax}\left(\frac{QK^\top}{\sqrt{d_k}} \diamond \mathbf{M}\right)
> $$
> where $\diamond$ is addition $(+)$ or element-wise multiplication $\odot$. LCAAM supports both self-attention $\mathbf{M} \in \mathbb{R}^{B \times N \times N}$ and cross-attention $\mathbf{M} \in \mathbb{R}^{B \times N_q \times N_k}$ setups and extends across multiple Transformer layers with layer-specific masks $\mathbf{M}_{l}$. This design integrates hierarchical and contextual learning, adapting dynamically to token importance at each layer.
>
> LCAAM scales to multi-layer Transformers by learning distinct masks $\mathbf{M}_l$ for each layer, adapting dynamically to the evolving representations as the input passes through the stack. This hierarchical design captures contextual and intermodal relationships across layers, providing nuanced control over attention distribution and enhancing model expressiveness.
>
> 2. **LCAAM shows limited gains in single-modality tasks. It would be useful to investigate why this is the case and whether adjustments could improve its effectiveness in these scenario:**
>
> As demonstrated in **Table 5**, **LCAAM's performance significantly improves with larger datasets and longer**, more complex input sequences. In single-modality setups, increasing dataset size enhances LCAAM’s ability to capture nuanced relationships, while in multi-modality settings, complex and lengthy inputs, such as the audiovisual tokens used in the Audio Description Task further boost performance. Therefore, training on extensive datasets and utilizing longer and more complex input sequences are promising strategies to maximize LCAAM's effectiveness across the single modality applications.
> The improvements in single-modality tasks are modest due to the simpler nature of processing a single input type, which underutilizes LCAAM’s dynamic masking. Future work could focus on amplifying LCAAM’s impact in single-modality tasks, possibly by incorporating additional contextual information or hybrid strategies combining insights from multiple modalities.

---

> ### Author Response · Authors · 2024-11-22
> **Response to the Reviewer b6MN (Part 2)**
>
> ### Questions
>
> 1. **Can you delve deeper into why LCAAM improves performance? Specifically, how does the mask influence the attention distribution, and what's the theoretical basis for its effectiveness?:**
>
> Evidence of LCAAM's capabilities is presented in **Figures S5, S6, S7, and S9** in the Appendix. For example, Figure S6 illustrates the masking values assigned to each token, where the x-axis represents the sequence of concatenated tokens: tokens numbered from 0 to 24 are visual tokens, and tokens from 25 to 49 are audio tokens. For visual token 1, LCAAM assigns values greater than 0.35 to tokens the visual tokens (0 to 24) indicating strong correlations among these visual elements, as depicted by the yellow-green-colored cells in the heatmap. While there is some correlation with audio tokens (25 to 49), these values are generally lower.
> Conversely, for audio token 1, LCAAM assigns higher values to the initial visual tokens and lower values to the later visual tokens. This reflects the static nature of the visual information—a backyard scene with minimal dynamic changes—while the audio tokens receive varying degrees of attention. Notably, the last audio tokens (e.g., Audio Tokens 15 to 25) correspond to indoor sounds, indicating a scene transition from an outdoor to an indoor setting. Consequently, LCAAM assigns values less than 0.35 to these tokens in its masking, interpreting them as less important and less related to the predominantly outdoor visual and audio tokens. **Figure S5** further suggests that LCAAM effectively prunes redundant connections within the attention mechanism, potentially leading to more computationally efficient model training without sacrificing performance. The observed sparsity aligns with recent trends in neural network optimization and opens avenues for further research into the interpretability and efficiency of attention-based models. These findings underscore LCAAM's potential as a promising approach for enhancing the scalability and resource utilization of Transformer-based architectures.
> Our theoretical basis is rooted in empirical observations detailed in Lines 037 to 048. Consider a movie scene represented by video and audio tokens. While these tokens naturally align in time, each one can be associated with any other token present in the scene, as shown in Figure 1(a). Although the self-attention module is effective for computing local associations between tokens, we have observed several drawbacks in the current attention mechanism, especially when tokens originate from diverse modalities. Firstly, different modalities introduce varying granularities of information, leading to potential challenges. Each token in one modality may be associated with multiple tokens in another modality. Such associations can extend beyond one-to-one correspondences, forming connections between sub-sequences of tokens in each modality. Additionally, as mentioned in Lines 049 to 053, other empirical studies have highlighted similar issues. However, this problem hasn't been thoroughly explored in the computer vision field, which serves as our motivation for this research.
>
> 2. **How does LCAAM stack up against other dynamic attention mechanisms like sparse attention or adaptive attention spans?
> Including direct comparisons would clarify its advantages:**
>
> LCAAM demonstrates clear advantages over other dynamic attention mechanisms, such as sparse attentions, particularly in handling complex and dynamic input sequences. Sparse attention, as exemplified by SwinBERT, is limited by its static learnable mask, which fails to adapt to the fluid and intricate nature of sequences like those in MAD-v2 **(Table 2)** . This rigidity leads to an inability to capture essential contextual relationships during rapid transitions, frequent shot changes, and multimodal interactions, as evidenced by its CIDEr score of 9.72. In contrast, LCAAM achieves significantly higher scores **(16.58 in the single-layer configuration and 17.11 in the multi-layer configuration)**, showcasing its capacity to dynamically adjust attention based on the data distribution. Unlike sparse attention, LCAAM’s dynamic framework allows it to seamlessly adapt to the shifting demands of MAD-v2’s complex scenarios, ensuring that critical contextual information is retained and effectively integrated across frames. Compared to adaptive attention spans, which focus on varying the scope of attention over fixed intervals, LCAAM offers a more granular and flexible mechanism, dynamically modulating attention across multiple modalities and temporal transitions. This adaptability enables LCAAM to better capture intricate relationships and maintain coherence across frames, providing a distinct edge over competing attention mechanisms in environments that demand high responsiveness and contextual awareness.

---

> ### Author Response · Authors · 2024-11-22
> **Response to the Reviewer b6MN (Part 3)**
>
> ### Questions
>
> 3. **Could you provide more insights into the computational overhead, such as actual latency measurements or memory usage in practical settings?:**
>
> **Table 4** presents an analysis of computational overhead, focusing on latency measurements for three different models. The **Baseline** model, with 6.93 billion parameters, achieves a latency of 87.57 ms, striking a balance between moderate performance (R-L = 12.92, C = 15.46) and efficient computational requirements (3.39 TFLOPs, 1.69 TMACs). The **Same Number of Parameters** variant retains a similar parameter count (7.072 billion) but exhibits slightly higher computational demands (3.45 TFLOPs, 1.80 TMACs), resulting in increased latency (89.12 ms) and reduced performance (R-L = 11.23, C = 12.87).  The **Multi-Layer LCAAM** model, with the same parameter count and a minor increase in MACs (1.71 TMACs), demonstrates significant performance improvements (R-L = 14.28, C = 17.11) while incurring only a marginal latency increase of 1.03 ms compared to the baseline (88.60 ms). These findings highlight how the Multi-Layer LCAAM architecture enhances resource utilization efficiently without a notable impact on response time.

---

### Public Comment · ~Siyuan_HUANG7 · 2024-11-21
**Applying Masks to Positive and Negative Un-Softmaxed Attention Weights**

Hello,

Thank you for your intriguing work. I have also been contemplating effective ways to incorporate masks into attention mechanisms recently. In Equation 5 of your paper, you mentioned that the "un-softmaxed attention weights" can be element-wise multiplied directly with a Mask. Regarding this, I have a question:

Does the Mask matrix values represent relative importance? (For example, in Figures S6 and S8, the mask values range between 0 and 1. Does this mean 0 tends towards unimportant, while 1 leans towards important?) Suppose the algorithm learns a perfect 0-1 mask, then using element-wise multiplication might pose an issue, as the un-softmaxed attention weights could include negative values, but multiplying negatives by 0 could paradoxically increase them. For instance, consider two token sections with original attention weights of -5 and -3 respectively. If the learned mask deems the first section unimportant and assigns a 0 mask value, and a 1 mask value to the second section, after element-wise multiplication, the attention weight of the first section becomes more prominent.

In summary, the issue is that the scaling effect of the mask when multiplied by positive and negative "un-softmaxed attention weights" is different, making it odd to apply the mask in this manner. I raise this issue because I have also been trying something similar in the NLP field recently. Interestingly, this "strange" operation of multiplying the mask with "un-softmaxed attention weights" has shown impressive performance in my experiments. I am curious about the underlying reasons.

---

> ### Author Response · Authors · 2024-11-22
> **Response to Siyuan HUANG**
>
> Thank you for your insightful question and for delving deeply into our approach.
>
> In our method, the Learnable Context-Aware Attention Mask (LCAAM) generates a mask $M$ that acts as a soft modulator for attention scores, enhancing their contextual relevance before softmax is applied:
>
> $$
> \text{Masked Scores} = \text{Scores} \odot M
> $$
>
> The mask values encode relative importance, where higher values emphasize strong relevance and lower values indicate lesser importance. The behavior you described with negative un-softmaxed scores and their scaling by the mask is addressed implicitly through the interaction with **the softmax function**. **Softmax computes the final attention distribution based on relative differences between scores**:
>
>
> $$
> \text{Attention Weights}_i = \frac{e^{z_i}}{\sum_j e^{z_j}}
> $$
>
> Since softmax depends on exponential transformations, the scaling effect introduced by $M$ preserves relative importance even for negative values, as all scores are exponentiated before normalization. Consequently, if certain unimportant tokens are masked with values near zero, their influence in the final attention distribution diminishes\. Similarly, tokens deemed important by the mask (with values close to 1) retain their relative prominence.
>
> Your observation regarding potential paradoxical amplification when negative scores interact with a zero-valued mask is mitigated in our experiments. The mask's **learnable nature** ensures its values adapt during training to effectively align with the model's overall objectives.

---

> > ### Public Comment · ~Siyuan_HUANG7 · 2024-11-24
> >
> > Thank you for your detailed response. I agree that LCAAM manages to preserve the relative magnitudes through the softmax mechanism, even with negative values. This indeed seems like a plausible reason why this mask method is effective. I appreciate your work, and I hope your paper gets the acceptance it deserves. :-)

---

> > > ### Author Response · Authors · 2024-11-27
> > > **Response to the public comment**
> > >
> > > Thank you for your interest in the paper and your kind support. I’m glad the explanation was helpful.

---

### Meta-Review · Area_Chair_Htz6 · 2024-12-19

**Metareview:**

This work introduces a new attention strategy, i.e., Learnable Context-Aware Attention Mask (LCAAM), to weight tokens according to importance for transformer models. Main concerns of reviewers come from limited gains in single-modality tasks, handling arbitrary input lengths, effectiveness of MLP layers for self-attention etc. After the discussion, some concerns are not fully addressed and more concerns are proposed by Reviewer sa9g. AC encourages authors to incorporate suggestions from all reviewers to polish and submit the work for the next venue.

**Additional Comments On Reviewer Discussion:**

During rebuttal, concerns from Reviewer MBg4 had been addressed but Reviewer sa9g found more concerns, e.g., fair comparison, and had the score of reject after the extensive discussion.

---

### Decision · Program_Chairs · 2025-01-22

Reject